# RNA exosome mutations in pontocerebellar hypoplasia alter ribosome biogenesis and p53 levels

Juliane S Müller[1,2,*], David T Burns[1,3,*], Helen Griffin[1,*] , Graeme R Wells[3], Romance A Zendah[3] , Benjamin Munro[1,2] , Claudia Schneider[3,*] , Rita Horvath[1,2,*]

The RNA exosome is a ubiquitously expressed complex of nine core proteins (EXOSC1-9) and associated nucleases responsible for RNA processing and degradation. Mutations in *EXOSC3*, *EXOSC8*, *EXOSC9*, and the exosome cofactor *RBM7* cause ponto-cerebellar hypoplasia and motor neuronopathy. We investigated the consequences of exosome mutations on RNA metabolism and cellular survival in zebrafish and human cell models. We observed that levels of mRNAs encoding p53 and ribosome biogenesis factors are increased in zebrafish lines with homozygous muta-tions of *exosc8* or *exosc9*, respectively. Consistent with higher p53 levels, mutant zebrafish have a reduced head size, smaller brain, and cerebellum caused by an increased number of apoptotic cells during development. Down-regulation of *EXOSC8* and *EXOSC9* in human cells leads to p53 protein stabilisation and G2/M cell cycle arrest. Increased p53 transcript levels were also observed in muscle samples from patients with *EXOSC9* mutations. Our work provides explanation for the pathogenesis of exosome-related disorders and highlights the link between exosome function, ribosome biogenesis, and p53-dependent signalling. We suggest that exosome-related disorders could be classified as ribosomopathies.

## Introduction

The core of the eukaryotic RNA exosome is composed of nine protein subunits (EXOSC1–EXOSC9 in *Metazoa*) and forms a two-layered ring, through which substrate RNAs can pass (Mitchell et al, 1997; Liu et al, 2006; Bonneau et al, 2009). The upper ring, the so-called cap of the exosome, is composed of EXOSC1-3 and the lower hexameric ring, or barrel of the exosome, is formed by EXOSC4-9. Catalytic activity of the human exosome is provided by the loosely associated EXOSC10/RRP6, DIS3/RRP44, or DIS3-like ribonucleases (Staals et al, 2010; Tomecki et al, 2010; Sloan et al, 2012). In the nucleus, the exosome processes and degrades a wide range of precursor RNAs, un-spliced pre-messenger RNAs, and cryptic transcripts (Allmang et al, 1999; Bousquet-Antonelli et al, 2000; Kadaba et al, 2004; Wyers et al, 2005; Gudipati et al, 2012; Sayani & Chanfreau, 2012; Schneider et al, 2012a; Schneider & Tollervey, 2013). Some reported that in the cytoplasm, the exosome degrades mRNAs that contain AU-rich elements (AREs) and RNAs that have evaded degradation in the nucleus (van Hoof et al, 2000; Mukherjee et al, 2002); however, no new evidence is available to corroborate this.

Recessive mutations in *EXOSC3*, *EXOSC8*, and *EXOSC9* have been associated with variable combinations of pontocerebellar hypoplasia (PCH) and spinal motor neuron dysfunction (Pontocerebellar hypoplasia type 1B, OMIM # 614678; Pontocerebellar hypoplasia type 1C, OMIM #616081; and Pontocerebellar hypoplasia type 1D, OMIM # 618065) and with central nervous system demyelination in patients carrying muta-tions in *EXOSC8* (Wan et al, 2012; Boczonadi et al, 2014; Muller et al, 2015; Burns et al, 2018; Morton et al, 2018). Bi-allelic *EXOSC2* mutations cause a usually milder and more complex genetic syndrome with short stature, dysmorphic features, myopia, retinitis pigmentosa, progressive senso-rineural hearing loss, hypothyroidism, premature ageing, and mild in-tellectual disability (Di Donato et al, 2016; Yang et al, 2019). A spinal muscular atrophy-like phenotype has been reported in a patient with a homozygous mutation in *RBM7*, another protein interacting with the exosome (Giunta et al, 2016).

Importantly, an intact RNA exosome is critical for the assembly of the ribosome, the essential RNA–protein machinery that synthesises all cellular proteins (Tafforeau et al, 2013; Sloan et al, 2013b, 2014; Gillespie et al, 2017). Defects in human ribosome production have been linked to more than 20 genetic diseases ("ribosomopathies") and can cause stabilisation of the tumour suppressor p53, leading to cell cycle arrest and apoptosis (Sloan et al, 2013a; Pelava et al, 2016; Aubert et al, 2018; Yang et al, 2018).

In patient fibroblasts, most reported core exosome–associated mu-tations result in reduced protein (Boczonadi et al, 2014; Burns et al, 2018). Reduction of single core exosome subunits impairs the assembly of the whole complex (Burns et al, 2018), which is likely to lead to the improper processing and degradation of RNA. Transcriptome-wide studies in-vestigated the effect of mutations in the human exosome–associated

---

[1]Wellcome Trust Centre for Mitochondrial Research, Institute of Genetic Medicine, Newcastle University, Newcastle upon Tyne, UK   [2]Department of Clinical Neurosciences, University of Cambridge School of Clinical Medicine, Cambridge, UK   [3]Biosciences Institute, Faculty of Medical Sciences, Newcastle University, Newcastle upon Tyne, UK

Correspondence: Rh732@medschl.cam.ac.uk
*Juliane S Müller, David T Burns, Helen Griffin, Claudia Schneider, and Rita Horvath contributed equally to this work

nucleases *DIS3* and *RRP6* (Szczepinska et al, 2015; Davidson et al, 2019), but it is still unclear which genes and species of RNA are most affected by mutations that are predicted to lead to a reduction in the core exosome complex in vivo. Discovering RNA types, which are perturbed by the abnormal function of the exosome could elucidate the mechanism behind core exosome–associated diseases and may identify potential novel molecular targets.

Gene knockdown via morpholino oligonucleotides targeting *exosc3*, *exosc8*, and *exosc9* has successfully recapitulated the patient phenotype in zebrafish. Knockdown of these genes in zebrafish resulted in cerebellar hypoplasia and muscle weakness likely due to failure of motor axons to migrate to neuromuscular junctions (Wan et al, 2012; Boczonadi et al, 2014; Burns et al, 2018). However, morpholino knockdown is transient and can sometimes produce off-target effects with varying severity of phenotypes (Kok et al, 2015; Stainier et al, 2015).

Previously, we studied transcript profiles in fibroblasts from patients with mutations in *EXOSC8, EXOSC9*, and the exosome cofactor *RBM7*, as well as in zebrafish where the expression of these genes was down-regulated via the injection of antisense morpholinos (Boczonadi et al, 2014; Giunta et al, 2016; Burns et al, 2018). In our previous work, we specifically selected poly(A)-tailed RNA for analysis by RNA sequencing and focused mainly on the cytoplasmic function of the exosome involving the degradation of AU-rich mRNA. However, we also detected a significant number of long non-coding RNAs among the differentially expressed transcripts (e.g., *HOTAIR*) (Giunta et al, 2016) and noticed cell line–specific differences in the poly(A)–tailed transcript profiles in patient fibroblasts depending on the gene defect. PCH subforms (with mutations in the tRNA-splicing endonuclease complex [TSEN], the cleavage and polyadenylation factor I subunit [*CLP1*], or the target of EGR1 protein 1 [*TOE1*]) have all been linked to changes in overall RNA metabolism, characterised by processing defects of small non-coding RNAs such as tRNAs and small nuclear RNAs (snRNAs) (Budde et al, 2008; Karaca et al, 2014; Schaffer et al, 2014; Breuss et al, 2016; Lardelli et al, 2017). The RNA exosome is known to be involved in processing and degradation of many, if not all, long and small non-coding RNA species in yeast and higher eukaryotes, which includes the precursors to the mature ribosomal RNAs (rRNAs) as well as small nuclear and nucleolar RNAs (sn[o]RNAs) (Gudipati et al, 2012; Schneider et al, 2012a; Szczepinska et al, 2015; Tomecki et al, 2017).

Here, we therefore use CRISPR/Cas9 to generate a stable line of zebrafish with mutations in *exosc8* and *exosc9*, allowing us to generate large numbers of homozygous mutants with a homogenous phenotype, and studied the impact of exosome disruption on zebrafish development, RNA metabolism, and cellular survival in the mutant fish. In parallel, we also performed siRNA-mediated down-regulation of EXOSC8 and EXOSC9 in human cells to look at exosome deficiency in the mammalian system.

# Results

## RNA analysis in homozygous *exosc8* and *exosc9* mutant zebrafish reveals increased mRNA levels of factors involved in ribosome assembly and other RNA metabolism pathways

To obtain a stable in vivo model of PCH, we created mutant zebrafish lines using CRISPR/Cas9 (Fig 1). We selected zebrafish that were heterozygous for the frameshift mutation c.26_27del in *exosc8*

(NM_001002865:exon2:c.26_27del:p.E9fs) and c.198_208del in *exosc9* (NM_001006077:exon3:c.198_208del:p.K67fs) (Fig S1). Heterozygous zebrafish were crossed, and embryos were allowed to develop to 5 days post-fertilisation (dpf). At 5 dpf, ~25% of embryos in each clutch appeared to have smaller heads and eyes and were unable to inflate their swim bladder (Fig 1A). Homozygous mutant larvae died between 5 and 7 dpf; all further analyses were performed at 5 dpf before their death. This phenotype was consistent with what was previously seen with morpholino oligos targeting *exosc3*, *exosc8*, and *exosc9* (Wan et al, 2012; Boczonadi et al, 2014; Burns et al, 2018).

PCR and sequencing confirmed that the abnormal zebrafish were homozygous for *exosc8* (c.26_27del) and *exosc9* (c.198_208del) mutations, respectively (Fig S1A–D). Heterozygous mutation carriers appeared phenotypically normal (data not shown). Although we were not able to confirm the loss of exosc8 or exosc9 on a protein level in the homozygous mutants because of the lack of a suitable antibody, we assume that very little or no functional protein was synthesised in the homozygous animals.

We performed RNAseq analysis on total RNA to see if there were abnormalities in the RNA profiles of the homozygous mutant zebrafish. The RNAseq analysis revealed that there was little reduction in the levels of the *exosc8* mRNA in the *exosc8* mutants (log$_2$fold change in RNAseq analysis of –0.052), which was surprising given that the c.26_27del mutation would be predicted to trigger mRNA degradation via nonsense-mediated decay, but it might be that exosome deficiency leads to reduced nonsense-mediated decay. The *exosc9* mRNA, however, was reduced to about 50% of the wild-type levels in the *exosc9* mutants in the RNAseq analysis (log$_2$fold change = –0.865). A reduction was also indicated by RT-PCR (non-quantitative PCR, Fig S1E).

Published RNAseq data from 3-dpf-old zebrafish embryos were available for comparison, where the expression of *exosc3* was down-regulated via antisense morpholino oligonucleotide injection targeting the start codon of *exosc3* and blocking *exosc3* translation (Francois-Moutal et al, 2018). 185 protein-coding transcripts were significantly increased in both mutants and the *exosc3* morphants, including eight transcripts that showed greater than twofold increase in steady-state levels in all mutants. The *exosc3* morphants, however, had a more severe phenotype than our mutants, possibly because of the fact that translation-blocking morpholinos also target the maternally deposited mRNA, not only the RNA transcribed by the embryo. This might have reduced the number of common transcripts. Somewhat surprisingly, given that the RNA exosome is an RNA decay/processing machine, 333 transcripts were significantly decreased with 26 transcripts showing a greater than twofold decrease in all mutants than in control embryos. This is mainly driven by a higher number of decreased transcripts in the *exosc8* mutant fish (Fig S2). Exosome dysfunction might cause a negative feedback loop on transcription levels.

Over-representation analysis of protein-coding genes that were significantly increased in homozygous mutant *exosc8* and *exosc9* zebrafish embryos and the *exosc3* morphants compared with controls showed higher levels of RNAs in gene ontology (GO) terms associated with ribosome assembly, snoRNP/Small Cajal body specific RNP (scaRNP) complexes, and (t)RNA modifications (Figs 2A, S2, and Table S1). Although 70 of the 185 significantly increased

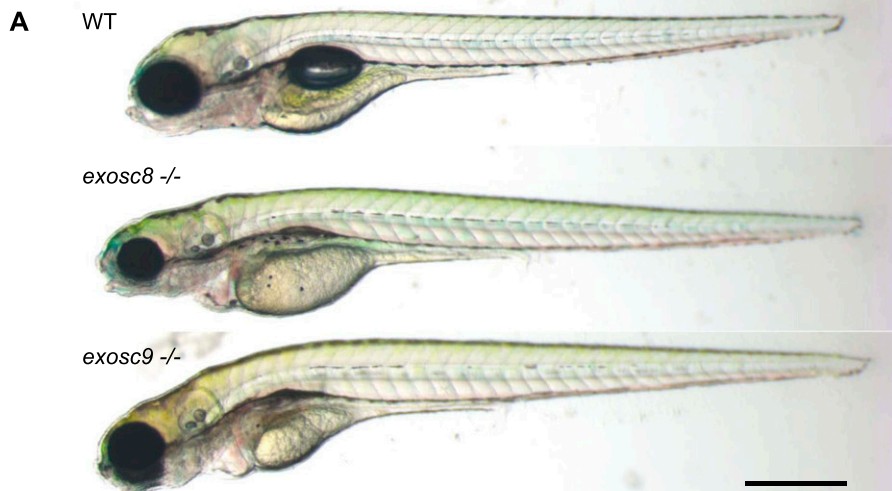

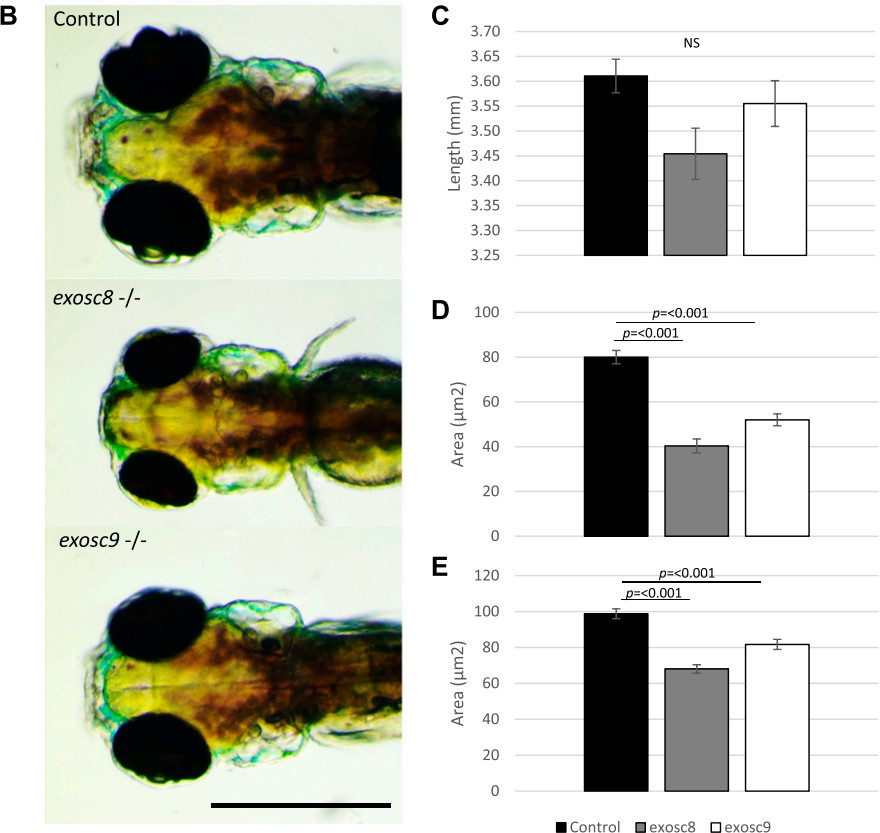

**Figure 1. *exosc8* and *exosc9* homozygous mutant zebrafish develop microcephaly.**
**(A)** Gross anatomy of 5-dpf wild-type, *exosc8* homozygous mutant and *exosc9* homozygous mutant zebrafish embryos; lateral view, anterior to the left. Scale bar: 500 μm. **(B)** Dorsal view of wild-type, *exosc8* homozygous mutant and *exosc9* homozygous mutant zebrafish embryos. Scale bar: 500 μm. **(C, D, E)** Standard length, (D) area of eyes, and (E) area of head in 5-dpf wild-type, *exosc8* homozygous mutant and *exosc9* homozygous mutant zebrafish embryos. 13 control, 8 *exosc8* (c.26_27del), and 5 *exosc9* (c.198_208del) homozygous larvae were measured for the quantification. Error bars represent the standard error (±SEM), and statistical analysis was performed using unpaired *t* tests (*exosc8* versus wt and *exosc9* versus wt, respectively). NS, not significant.

transcripts were associated with the general "RNA metabolic process" GO term, a remarkable 90% of those (63 out of the 70 transcripts, Table S2) were found to belong to the pathway of ribosome biogenesis and 51 of the 70 more precisely to the rRNA processing pathway (Fig 2B, http://pantherdb.org/).

In addition to changes in protein-coding transcripts, there was also a greater than twofold increase in normalised RNA read counts for 182 (9.0%) and 153 (7.6%) of 2,004 individual non-coding RNAs in the *exosc8* and *exosc9* homozygous mutant zebrafish, respectively,

compared with wild-type and heterozygous clutchmates (Fig 2C–E, selected non-coding RNAs represented in Fig S2). In contrast, 111 (5.5%) and 80 (4.0%) non-coding RNAs surprisingly showed a greater than twofold decrease than controls (Table S3). Overall, there was a significant increase in all non-coding RNA reads in *exosc8* embryos versus controls (t = 2.8653, *P*-value = 0.004), particularly RNAs belonging to the cytosolic tRNA, miscRNA (miscellaneous other RNAs that do not fall in any of the other categories), and small Cajal body–specific RNA (scaRNA) categories (Fig 2C). A

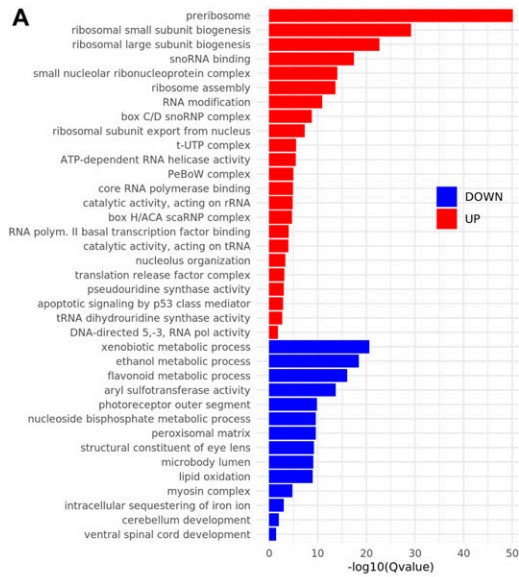

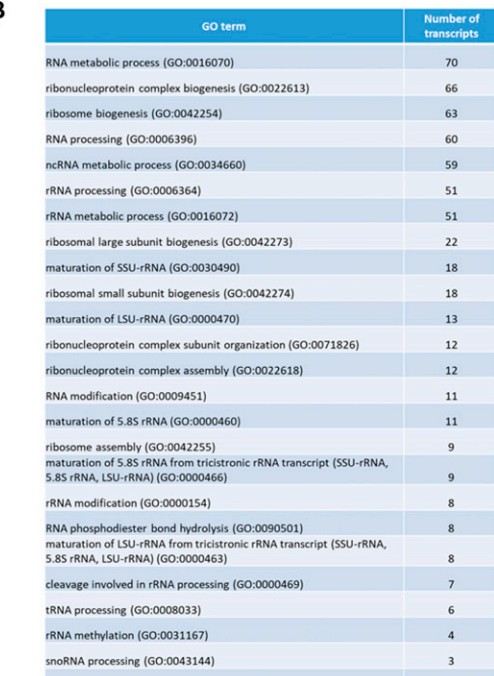

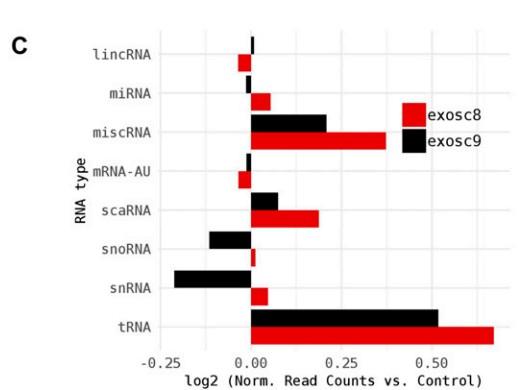

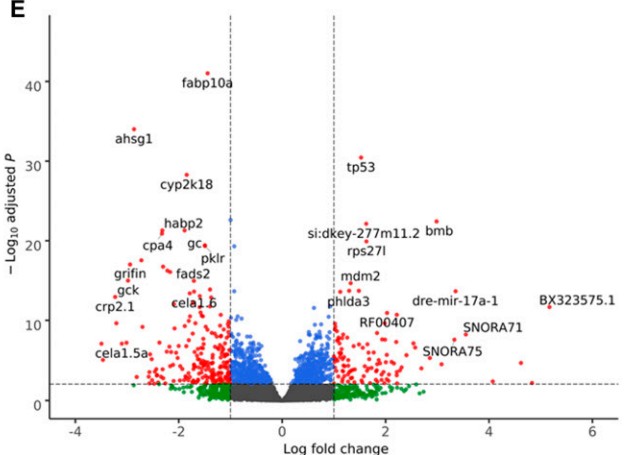

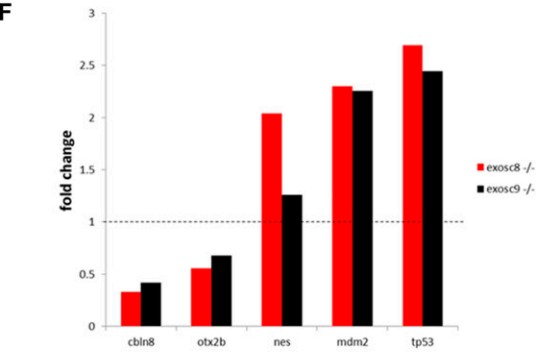

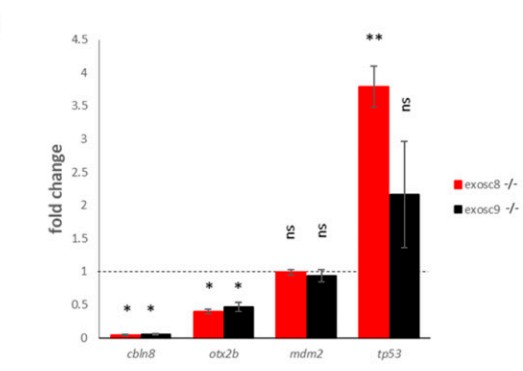

comparison of the per gene $\log_2$ ratios of non-coding read counts showed an overall positive correlation between *exosc8* and *exosc9* embryos at the individual gene level for all RNA categories (Fig 2C and D).

## Altered mRNA levels of apoptosis-related genes are consistent with an increased number of apoptotic cells in the head of *exosc8* and *exosc9* zebrafish mutants

In addition to pathways linked to ribosome biogenesis or RNA metabolism, there was a significant increase in RNA counts from six genes (*TP53*, *RPS27L*, *BAX*, *RRN3*, *MYBBP1A*, and *MDM2*) from the GO term "intrinsic apoptotic signalling by p53 class mediator" (GO: 0072332, qvalue = 0.001, Fig 2A). This was interesting, given that, although an intact exosome is essential for ribosome biogenesis, defects in ribosome production have been shown to cause stabilisation of the tumour suppressor p53, via inhibition of MDM2, leading to cell cycle arrest and apoptosis (Sloan et al, 2013a; Pelava et al, 2016; Aubert et al, 2018). The increase in *mdm2* transcript was 2.3 fold and 2.56 fold, respectively, in the *exosc8* and *exosc9* homozygous mutant embryos and in *p53* transcript 2.69 fold and 2.44 fold, respectively (Fig 2F). The increase in *p53* mRNA levels, but not in *mdm2* mRNA levels, was confirmed via qRT-PCR (Fig 2G).

Although there was no significant difference in the standard length of zebrafish larvae at 5 dpf (Fig 1C), *exosc8* and *exosc9* homozygous mutant zebrafish had significantly smaller eyes and heads (Fig 1B, D, and E). In addition, both the *exosc8* and *exosc9* homozygous embryos had a significantly smaller midbrain (Fig 3A and B) and cerebellum (Fig 3A and C) than WT and heterozygous clutchmates, fewer motor axon branches (Fig S3) at 5 dpf, and thinner tails with disorganized skeletal muscle fibres (Fig S4).

As the *exosc8* and *exosc9* homozygous mutants had noticeably smaller heads and eyes, cerebellar atrophy, and significant up-regulation of transcripts from the p53-dependent signalling pathway, we performed acridine orange staining to see if we can detect an increase in apoptosis in the mutant zebrafish embryos. Indeed, in the heads of the *exosc8* and *exosc9* homozygous mutant zebrafish embryos, there appeared to be a pronounced increase in apoptosis at 48 hours post-fertilisation (hpf) (Fig 4A); quantification of apoptotic cells (=acridine orange positive spots) showed a significant increase in homozygous mutants compared with wild-type and heterozygous embryos (Fig 4B). This increase in apoptosis would lead to a reduction in the number of cells in the head and could be the cause of the observed microcephaly and cerebellar hypoplasia or atrophy observed in the *exosc8* and *exosc9* homozygous mutant embryos at 5 dpf (Figs 1 and 3).

We had access to and have previously published RNAseq data from muscle biopsies of two patients with *EXOSC9* mutations (Burns et al, 2018). Analysis showed significant changes in expression of a large number of genes: 2,778 genes were significantly up-regulated and 2,864 down-regulated in both individuals (Table S4), among them transcripts originating from genes linked to motor neuronopathy, familiar amyotrophic lateral sclerosis, or distal arthrogryposis. In addition, several pathways linked to p53 (such as DNA damage response, signal transduction by p53 class mediator, and intrinsic apoptotic signalling pathway by p53 class mediator) and genes encoding components of the large ribosomal subunit were also significantly increased (Table S4). This confirms that core exosome deficiency directly affects p53-dependent signalling also in patient tissues.

## RNAi-mediated down-regulation of core exosome subunits in cultured human cells leads to a significant increase in p53 protein levels and a G2/M cell cycle arrest

To investigate the effect of core exosome disruption on the p53-dependent signalling pathway in more detail and on a protein level, HCT116 cells (a human colorectal cancer cell line) were treated with a siRNA targeting the *EXOSC8* or *EXOSC9* mRNA or a control siRNA targeting firefly luciferase. Western blot analysis of total protein extract from siRNA-treated cells confirmed EXOSC8 or EXOSC9 protein depletion upon treatment with their target siRNA (Fig 5A, left panel). In addition, siRNA-mediated depletion of *EXOSC8* or *EXOSC9* also significantly reduced EXOSC9 or EXOSC8 protein levels, respectively. This finding is in agreement with previous studies, in which reduction of single core exosome subunits impaired the assembly of the whole complex (Burns et al, 2018). Importantly, down-regulation of both core exosome subunits by RNAi resulted in a ~1.7 fold (*EXOSC8*) or ~2.5 fold (*EXOSC9*) statistically significant increase in p53 protein levels (Fig 5A, right panel). These data are consistent with the high p53 mRNA levels observed in *exosc8*$^{-/-}$ and *exosc9*$^{-/-}$ homozygous zebrafish (Fig 2F and G) and could reflect increased stability and/or higher translation rate of the p53 mRNA in HCT116 cells in the absence of an intact core exosome.

However, it is also important to note that RNAi-mediated depletion of *EXOSC8* and *EXOSC9* has previously been shown to cause a defect in the processing of rRNA in cultured cells (Tafforeau et al, 2013), and p53 protein stability is directly linked to ribosome production (reviewed in Pelava et al [2016]). Defects in ribosome biogenesis cause accumulation of the 5S RNP, a large subunit ribosome assembly intermediate composed of the 5S rRNA and two ribosomal proteins (RPL5 and RPL11). The 5S RNP regulates p53 protein levels by binding to and inhibition of the E3 ubiquitin protein ligase MDM2, which targets p53 for degradation (Sloan et al, 2013b).

**Figure 2. *exosc8* and *exosc9* homozygous mutant zebrafish have increased non-coding RNA expression and differences in gene expression.**
Summary of RNAseq performed on 5-dpf wild-type, *exosc8* homozygous mutant and *exosc9* homozygous mutant zebrafish embryos. **(A)** Gene set over-representation analysis of RNAs that were significantly increased or decreased in both homozygous mutant *exosc8* and *exosc9* zebrafish embryos compared with controls. **(B)** Selected GO terms and number of transcripts associated with them among the increased protein-coding RNAs in the mutant zebrafish lines. GO term analysis was performed on http://pantherdb.org/. **(C)** $\log_2$ ratios of cumulative, normalised RNA read counts for different non-coding RNA types in *exosc8* or *exosc9* homozygous mutant embryos versus controls. **(D)** Correlation of *exosc8* and *exosc9* $\log_2$ ratios of cumulative, normalised read counts for individual non-coding RNA genes. **(E)** Volcano plot showing the differential expression of transcripts in *exosc8* and *exosc9* homozygous mutants versus controls, with statistical significance (*P*-value) on the y-axis versus the magnitude of change (fold change) on the x-axis. **(F)** Selected mRNAs where increased levels were observed in *eoxsc8* and *exosc9* homozygous mutant zebrafish. **(F, G)** Validation of changes in gene expression via qRT-PCR for the mRNAs represented in (F). Error bars represent the standard error (±SEM), and statistical analysis was performed using unpaired *t* tests (*exosc8* versus wt and *exosc9* versus wt, respectively). Values have been normalised to a housekeeping gene and to wt fish; wt level has been set to one and is represented by the dotted line. *\*P*-value < 0.05, *\*\*P*-value < 0.01, NS, not significant.

 Life Science Alliance

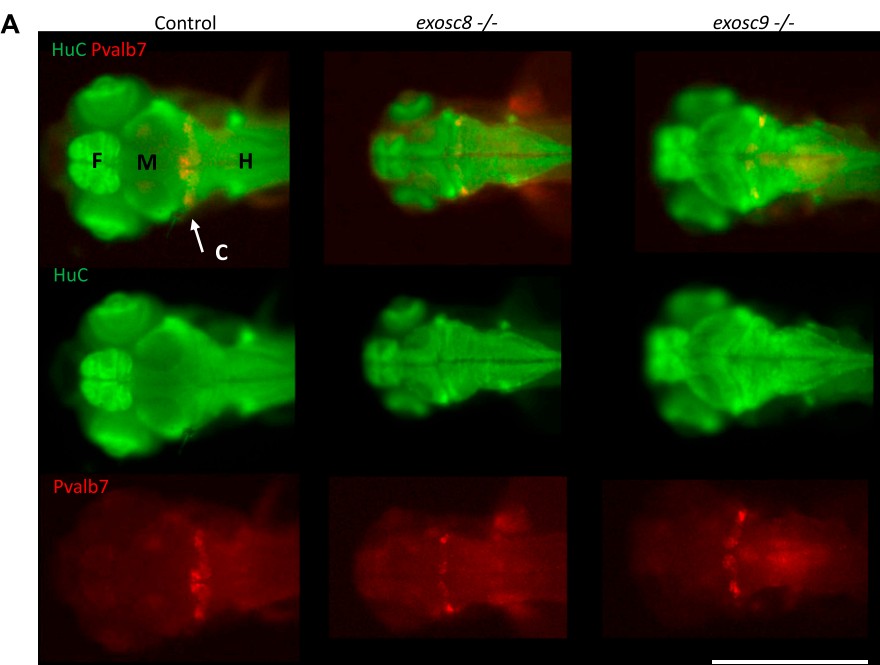

**Figure 3. *exosc8* and *exosc9* homozygous mutant zebrafish develop cerebellar atrophy.**
**(A)** Immunofluorescence in 5-dpf wild-type, *exosc8* homozygous mutant and *exosc9* homozygous mutant zebrafish embryos with antibodies raised against HuC (green) and Pvalb7 (red) Pvalb7 is a marker for Purkinje cells in the cerebellum, HuC is an early neuronal marker. Scale bar: 500 μm. **(B, C)** Quantification of (B) HuC-positive area and (C) Pvalb7-positive area in 5-dpf wild-type, *exosc8* homozygous mutant and *exosc9* homozygous mutant zebrafish embryos. Three control, six *exosc8* (c.26_27del), and six *exosc9* (c.198_208del) homozygous larvae were measured for the HuC quantification. 6 control, 7 *exosc8* (c.26_27del), and 11 *exosc9* (c.198_208del) homozygous larvae were measured for the Pvalb7 quantification. Error bars represent the standard error (±SEM) and statistical analysis was performed using unpaired *t* tests (*exosc8* versus wt and *exosc9* versus wt, respectively). Labelling: C, cerebellum; F, forebrain; H, hindbrain and spinal cord; M, midbrain.

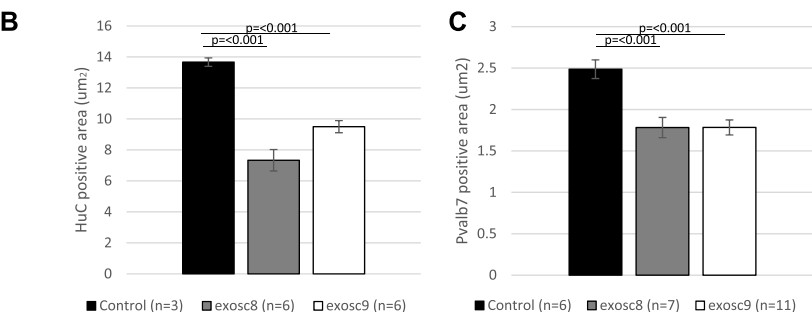

We therefore considered that stabilisation of the p53 protein caused by a defect in ribosome production may also strongly contribute to the higher p53 protein levels seen upon EXOSC9 knockdown (Fig 5A), likely in addition to the proposed increased stability of the p53 mRNA. To test this hypothesis, siRNA-mediated depletion of EXOSC9 was performed in conjunction with depletion of the 5S RNP component RPL5. Depletion of both EXOSC9 and RPL5 in HCT116 cells did not lead to increased levels of p53 protein compared with treatment with a control siRNA, and p53 protein levels were significantly reduced upon double depletion compared with depletion of EXOSC9 alone (Fig 5B). Given that RPL5 depletion is not expected to directly affect p53 mRNA stability, these data indicate a 5S RNP-dependent stabilisation of the p53 protein due to a ribosome biogenesis defect occurring upon EXOSC9 depletion in HCT116 cells.

To further determine whether siRNA-mediated down-regulation of EXOSC9 and the resultant increase in p53 protein levels affect the cell cycle, propidium iodide (PI) DNA staining of HCT116 cells was performed in combination with flow cytometry (Fig 5C and D). Cells were treated with a control siRNA or a siRNA targeting EXOSC9 mRNA for ~60 h before staining and FACS analysis (using a FACS Canto II flow cytometer and FACSDIVA software [BD Biosciences]). Treatment with 4 nM actinomycin D

(ActD), a chemotherapeutic agent known to block rRNA synthesis by inhibiting RNA polymerase I, was used as a positive control, as it has been shown previously to cause increased p53 levels (~4 fold) as well as a G1 and G2/M cell cycle arrest (Fumagalli et al, 2012; Sloan et al, 2013a).

Analysis of the FACS data showed that treatment with 4 nM ActD for 18 h caused a significant G1 and G2/M cell cycle arrest, as well as a significant reduction in the S phase, compared with untreated control HCT116 cells (Fig 5C and D). Depletion of EXOSC9 also caused a significant G2/M cell cycle arrest together with a decrease of cells in the G0/G1 phase, consistent with its effect on p53 protein levels (Fig 5C and D) but did not change the distribution of cells in the S phase of the cell cycle. The phenotype observed upon *EXOSC9* depletion is therefore reminiscent of the nucleolar stress response (Yang et al, 2018) and similar, but milder than, the cell cycle defect observed upon treatment with ActD.

### Transcriptome analysis in exosome patient cells

As HCT116 cells are a cancer cell line, we also wanted to study changes on transcript levels and the p53 pathway in neuronal cells

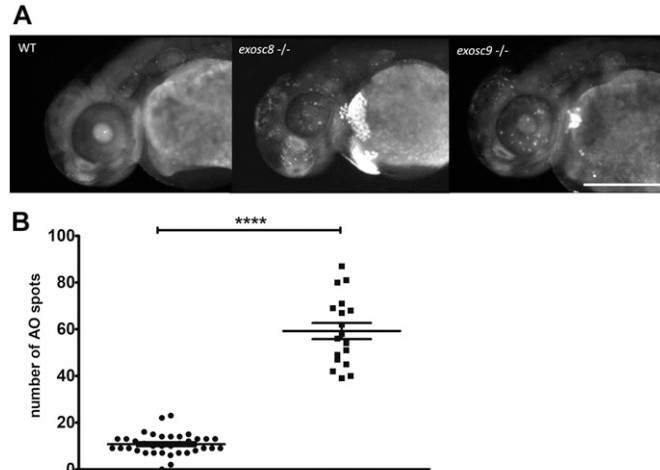

**Figure 4. *exosc8* and *exosc9* homozygous mutant zebrafish have increased apoptosis.**
**(A)** Acridine orange staining performed on 48-hpf wild-type, *exosc8* homozygous mutant and *exosc9* homozygous mutant zebrafish embryos. Representative images of each category are shown here. Four different clutches of offspring with 24 embryos each have been analysed; the acridine orange staining has been performed and evaluated first (without knowledge of the genotype), followed by genotyping of all embryos that were stained. Scale bar: 500 *μ*m. **(B)** Quantification of acridine orange positive spots. Spots were counted in 36 images of wild-type or heterozygous embryos and 18 images of homozygous mutant embryos. Bright spots in the yolk sac were not counted. Error bars represent the standard error (±SEM) and statistical analysis was performed using the unpaired *t* test. ****$P < 0.0001$.

originating from patients with exosome mutations. We collected fibroblasts from patients carrying the *EXOSC3* mutation p.Gly31Ala, the *EXOSC8* mutation p.Ala2Val, and the *EXOSC9* mutation p.Leu14Pro, respectively. We reprogrammed the fibroblasts into induced neuronal progenitor cells (iNPCs), thus creating a cell culture model which is more closely related to the affected nervous system (Fig S5). All three iNPC lines showed a small overall increase in total non-coding RNA sequence read counts compared with control iNPC lines, which were derived from fibroblasts taken from healthy, but genetically non-related individuals, with different distributions in the magnitude of RNA alteration per non-coding RNA category for each cell line (Figs 6A, S6, S7, and Table S5). The *EXOSC8* iNPCs showed an increase in total RNA read counts for all individual categories of non-coding RNAs, with the largest increase (+1.3 log$_2$) seen with snoRNAs, and also substantial increases in miRNA (+0.7 log$_2$), tRNA (+0.5 log$_2$), and snRNA (+0.3 log$_2$) categories. The *EXOSC3* cell line also showed an increase in snoRNAs (+0.6 log$_2$); however, this increase was absent from the *EXOSC9* cell line that overall showed a small decrease (−0.2 log$_2$) in snoRNAs. ScaRNA levels are also slightly increased in all three iNPC lines. The differential expression of the different categories of non-coding RNA matches our findings from the zebrafish mutants.

In the *EXOSC8* iNPCs, gene set over-representation analysis of the 321 and 220 RNAs that were significantly increased or decreased in the mutant iNPCs compared with controls revealed an up-regulation of the ER unfolded protein response and ER–Golgi intermediate compartment, as well as factors linked to intrinsic apoptotic signalling due to ER stress (Figs 6B, S8, and Table S6). A few differentially expressed genes (Fig 6C, individual plots Fig S8)

have been shown to play a role in development. We found increased levels of chordin (CHRD), a gene implicated in organising the early development of the mammalian head (Anderson et al, 2002), and decreased levels of calbindin 1 (CALB1), a gene associated with Cerebellar Disease and Purkinje Cells (Vig et al, 2012). The *EXOSC8* iNPCs also showed a slight overall increase in the levels of AU-rich element containing mRNAs (Fig 6A) in agreement with our previously published results (Boczonadi et al, 2014).

Unlike previously seen in zebrafish, none of the transcripts significantly increased in all three mutant iNPC lines related to ribosome assembly, rRNA processing, or p53 signalling (Table S6). Western blotting was performed on total protein from *EXOSC3*, *EXOSC8*, and *EXOSC9* patient and control iNPCs, but consistent with the observed TP53 transcript levels in the RNAseq datasets (Table S6), p53 protein levels were very variable in the different control cells, which made it impossible to draw major conclusions (Fig S5H).

## Differential gene expression in developmental pathways in *exosc8* and *exosc9* zebrafish mutants

The RNAseq differential expression analysis in zebrafish also showed significant dysregulation of genes that have previously been associated with disease or specific developmental pathways (Table S1). Down-regulated genes were mostly enriched in metabolic-related GO terms, with a few significantly down-regulated genes enriched in the terms cerebellum and ventral spinal cord development (Fig 2A and Table S1). In the *exosc8* and *exosc9* homozygous mutant embryos, there was a 0.33-fold and 0.42-fold reduction in *cbln8* (*cerebellin 8*) mRNA, respectively (Fig 2F). This was also confirmed via qRT-PCR (Fig 2G). Cerebellins are a family of adhesion molecules essential in the formation and function of synapses, especially in the Purkinje cells (Ito-Ishida et al, 2008; Uemura et al, 2010; Lee et al, 2012). Although nothing is known about the specific function of *cbln8*, its reduction could possibly impair the formation of synapses and influence the development of the cerebellum in the *exosc8* and *exosc9* mutant embryos.

In the *exosc8* and *exosc9* homozygous mutant embryos, there was a significant reduction in the expression of *otx2b* (orthodenticle homeobox 2, 0.56-fold, and 0.68-fold in the RNAseq data and 0.40 fold and 0.47 fold via qRT-PCR, respectively). OTX2 is a transcription factor expressed in the brain and has roles in midbrain–hindbrain boundary formation during development (Simeone et al, 1992; Ang et al, 1994; Kurokawa et al, 2012). Mutations in *OTX2* in humans have been associated with otocephaly and other craniofacial abnormalities that are reliably phenocopied in mouse and zebrafish models (Matsuo et al, 1995; Chassaing et al, 2012). Notably, ribosome biogenesis defects have also been linked to craniofacial abnormalities in humans and in animal models (Calo et al, 2018). To investigate the craniofacial structure in the *exosc8* and *exosc9* homozygous mutant zebrafish, we performed Alcian blue staining to visualise cartilage in the larvae. Alcian blue staining at 5 dpf illustrated abnormal cartilage development in the *exosc8* and *exosc9* homozygous mutant embryos (Fig 7A). Both the *exosc8* and *exosc9* homozygous mutants had a significantly shorter ceratohyal, a pharyngeal arch cartilage component, than wild-type and heterozygous clutchmates (Fig 7B). The angle of the ceratohyal was

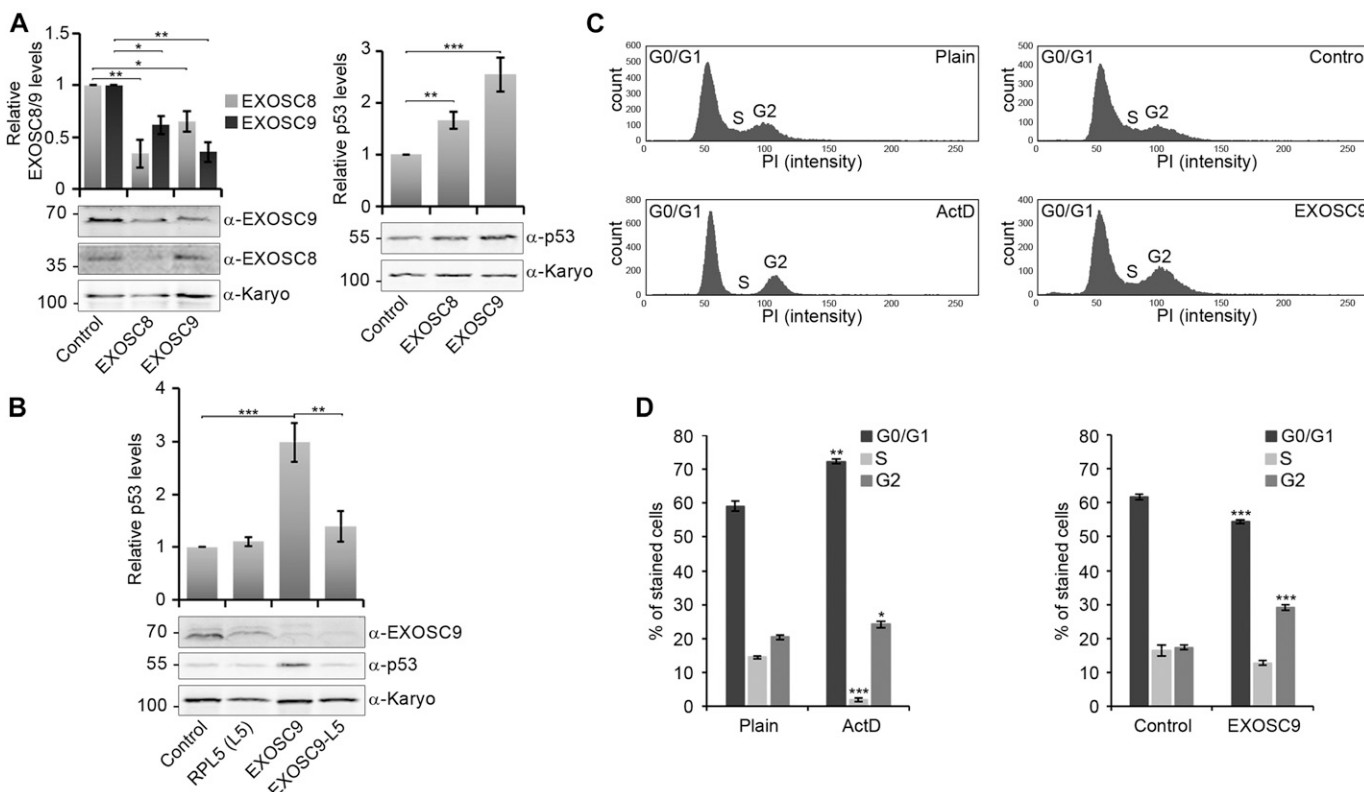

**Figure 5. RNAi-mediated core exosome subunit down-regulation in HCT116 cells leads to a 5S RNP-dependent increase in p53 protein levels and a G2/M cell cycle arrest.**
**(A)** HCT116 cells were treated with siRNAs targeting the EXOSC8 or EXOSC9 mRNA or a control siRNA targeting firefly luciferase for 60 h. Protein levels were analysed by immunoblotting using antibodies specific for EXOSC8 or EXOSC9 (left panel), p53 (right panel) or karyopherin as loading control (both panels) and at least three independent repeat experiments were quantified using ImageQuant (GE Healthcare) software. **(B)** HCT116 cells were treated with siRNAs targeting EXOSC9 and/or core 5S RNP component RPL5 mRNA or a control siRNA targeting firefly luciferase for 60 h. Protein levels were analysed by immunoblotting using antibodies specific for EXOSC9 (upper panel), p53 (middle panel), or karyopherin as loading control (lower panel), and at least three independent repeat experiments were quantified using ImageQuant (GE Healthcare) software. **(C, D)** Cell cycle analysis was performed using the FACS Canto II flow cytometer and evaluated using FACSDIVA software. **(B)** Following actinomycin D (ActD, 4 nM, 18 h) or siRNA treatment (as in panel B), the cells were fixed using 70% ethanol, and DNA was stained using propidium iodide. **(C)** Representative profiles are shown in panel (C). **(D)** The graph in panel (D) shows the average percentage levels of G0/G1 (dark grey), S (light grey), or G2/M (grey) phase of at least three experimental repeats. **(A, B, D)** Error bars represent the standard error (±SEM), and statistical analysis was performed using unpaired t tests. Comparison of significance was performed against the respective control knockdown as indicated or against the plain HCT116 cells for ActD treatment. Absence of significance values indicates no significant differences to the control. *P-value < 0.05, **P-value < 0.01, ***P-value < 0.001.
Source data are available for this figure.

often hyperpolarized in the *exosc8* homozygous embryos, manifesting in a significantly greater angle when than clutchmates (Fig 7C). The distance between Meckel's cartilage, and the ceratohyal was significantly reduced in the *exosc9* homozygote embryos (Fig 7D). These are observations which have previously been observed in zebrafish models of craniofacial abnormalities (Noack Watt et al, 2016).

# Discussion

Here, we have established zebrafish models for PCH caused by mutations in *EXOSC8* and *EXOSC9*, which recapitulate the clinical phenotype observed in patients with underdevelopment of the cerebellum and neuromuscular defects. The head and brain size measurements might indicate that *exosc9* mutant fish have a slightly milder phenotype than *exosc8* mutants. Neuromuscular junctions, however, are slightly more affected in *exosc9* mutants (Fig S3), and both mutants die at the same developmental stage. Currently, it is difficult to compare disease severity in patients with *EXOSC8* mutations versus *EXOSC9* mutations as there is only a very small number of patients reported. It might be possible that *EXOSC9* patients have milder clinical symptoms than *EXOSC8* patients, but it may also depend on the exact mutation. For *EXOSC3* mutations, where many more patients have been reported, a clearer genotype–phenotype comparison has shown that certain *EXOSC3* mutations are associated with milder symptoms and better survival rates (Ivanov et al, 2018).

The *exosc8* and *exosc9* mutant fish also have a slightly milder phenotype than our previously published *exosc8* morphants (Boczonadi et al, 2014) and the *exosc9* morphants and crispants (Burns et al, 2018). For morpholino-mediated knockdown, studies have shown that the morphant phenotype is more severe than the

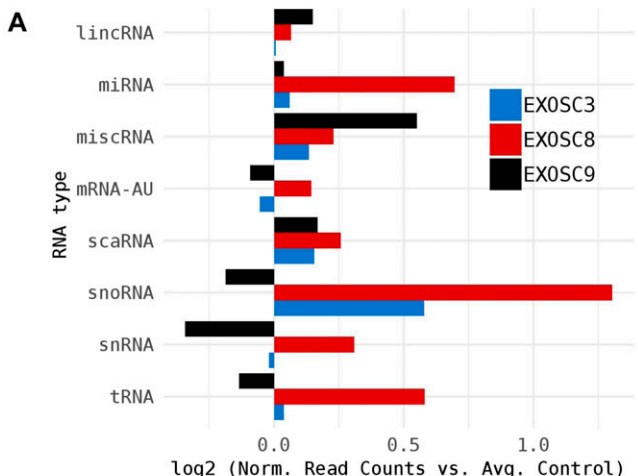

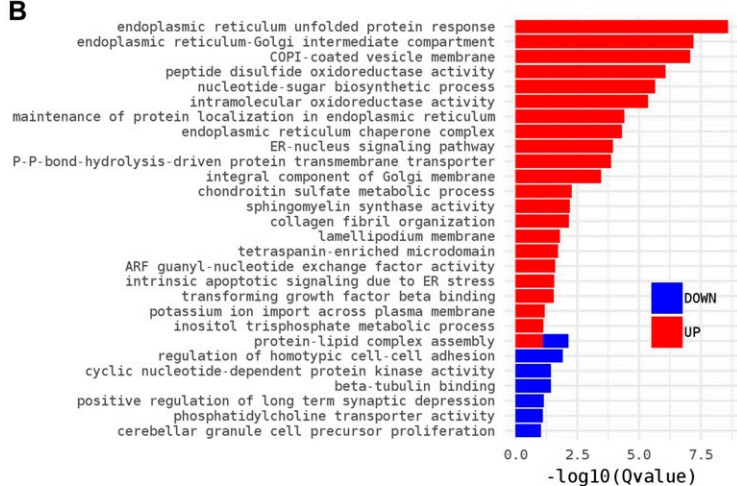

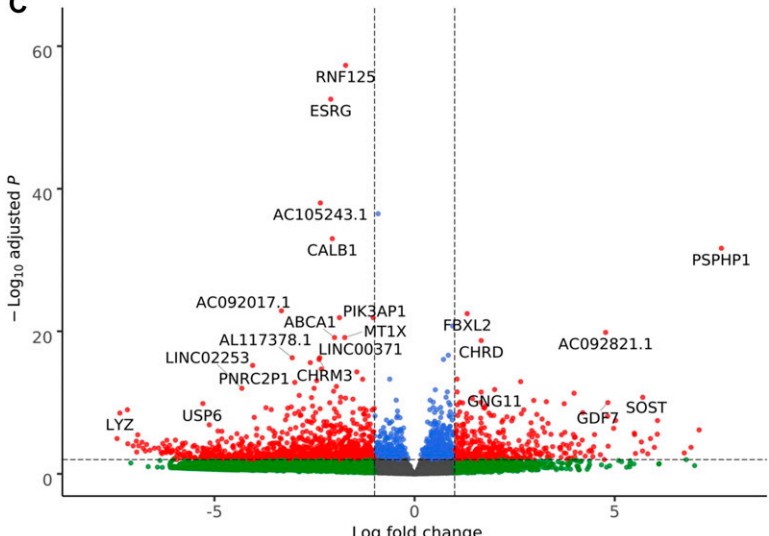

**Figure 6. *EXOSC3*, *EXOSC8*, and *EXOSC9* mutant–induced neuronal progenitor cells (iNPCs) have alterations in non-coding RNA expression that correlate with zebrafish mutants.**
Summary of RNAseq performed on iNPCs converted from fibroblasts carrying either *EXOSC3*, *EXOSC8*, or *EXOSC9* mutations or controls. **(A)** Log$_2$ ratios of cumulative, normalised RNA read counts for different non-coding RNA types in *EXOSC3*, *EXOSC8*, or *EXOSC9* mutant iNPCs versus the average of controls. **(B)** Gene set over-representation analysis of RNAs that showed a significant difference in expression between *EXOSC8* versus control iNPCs. **(C)** Volcano plot showing the differential expression of transcripts in *EXOSC3*, *EXOSC8*, and *EXOSC9* mutant iNPCs versus controls, with statistical significance (*P*-value) on the y-axis versus the magnitude of change (fold change) on the x-axis.

genetic mutant phenotype because there are some compensatory effects by other genes that become up-regulated and rescue the phenotype (Rossi et al, 2015). This might also be the case for the crispants, although it has not been studied yet. Some F0-injected crispants could show a more severe phenotype because of the injection process of RNA and Cas9. Furthermore, F0 fish might contain mutations in other genes caused by the guideRNA binding to other genomic sites; those are removed by

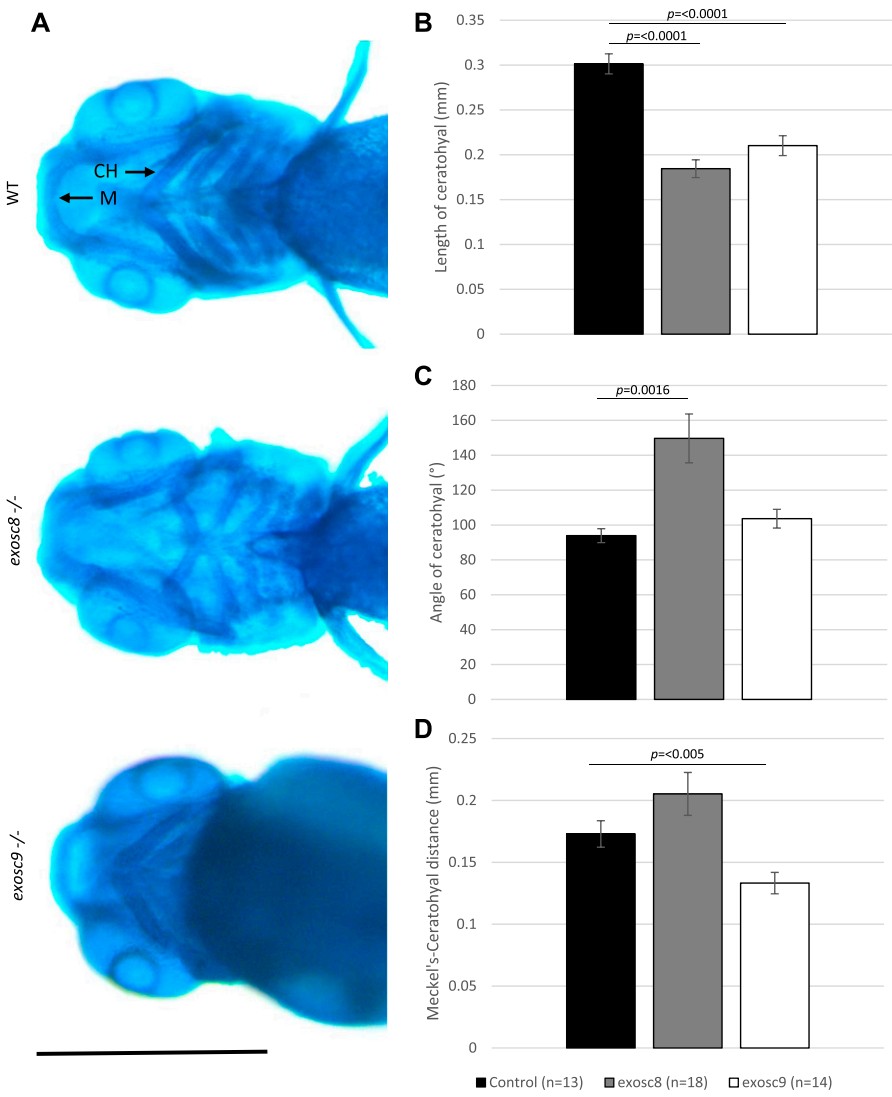

**Figure 7.   *exosc8* and *exosc9* homozygous mutant zebrafish develop craniofacial abnormalities.**
**(A)** Alcian blue staining in 5-dpf wild-type, *exosc8* homozygous mutant, and *exosc9* homozygous mutant zebrafish embryos. Scale bar: 500 *μ*m. **(B, C, D)** Length of ceratohyal, (C) angle of ceratohyal, and (D) distance from Meckel's cartilage to the ceratohyal in *exosc8* homozygous mutant and *exosc9* homozygous mutant zebrafish embryos. **(B, C, D)** 13 control, 18 *exosc8* (c.26_27del), and 14 *exosc9* (c.198_208del) homozygous larvae were measured for the quantification in (B, C, D). Error bars represent the standard error (±SEM), and statistical analysis was performed using unpaired *t* tests (*exosc8* versus wt and *exosc9* versus wt, respectively). The difference between mutant and wt is not significant where no *P*-value is given. Abbreviations: CH, ceratohyal; M, Meckel's cartilage.

outcrossing in later generations. However, we emphasize that also in our earlier study (Burns et al, 2018), *exosc9* crispants have a predominantly mild phenotype (only 16% have a moderate or severe phenotype).

RNAseq analysis performed in the *exosc8* and *exosc9* mutant zebrafish shows that levels of mRNAs encoding ribosome biogenesis factors and several non-coding RNA species are increased. In addition, we observed increased apoptosis of brain cells, correlating with higher p53 transcript levels in the mutants. The increase in p53 mRNA levels was also confirmed in muscle tissue from patients. In addition, p53 protein levels were found increased in a human cancer cell line when EXOSC8 or EXOSC9 were down-regulated using siRNA. Our analysis links exosome deficiency to defects in ribosome production, which, in turn, affect p53-dependent cellular signalling pathways. Inherited diseases caused by exosome subunit mutations could, therefore, be considered as ribosomopathies caused by ribosome dysfunction.

Work carried out by other groups showed that the knockdown of *EXOSC3* or the exosome-associated 3′-5′ exonucleases *DIS3* and *EXOSC10/RRP6* in mammalian cell lines had profound consequences on the levels of many different RNA types: Ablation of *EXOSC3* and *EXOSC10* in mouse embryonic stem (ES) cells (Pefanis et al, 2015) resulted in increased levels of long non-coding RNAs, antisense RNAs, and enhancer RNAs compared with wild-type control ES cell transcriptomes. Knockdown of *EXOSC3* by siRNA in mammalian cells was also shown to result in an enrichment of snoRNA precursors where introns had not been removed and of PROMoter uPstream Transcripts (PROMPTs) (Lubas et al, 2015). In agreement with these studies, we also detected increased levels of lincRNAs and snoRNAs in the iNPC from exosome patients. However, we did not observe clearly elevated levels of PROMPTs in our samples, except for a slight increase in read counts upstream of the transcription start site in the *EXOSC8* iNPC line, and to a lesser extent in *EXOSC3* and *EXOSC9* lines at 77 genes (Fig S9). The total number of reads in these regions was very low in general though, making the results unreliable. The lack of PROMPT accumulation might be due to different RNA isolation and analysis techniques compared with the published studies.

EXOSC3, EXOSC8, EXOSC9, EXOSC10, or DIS3 depletion causes the appearance of mis-processed 5.8S pre-rRNA species and accumulation of larger rRNA precursors (Tomecki et al, 2010; Lubas et al, 2011; Tafforeau et al, 2013; Sloan et al, 2013b, 2014; Kobylecki et al, 2018; Davidson et al, 2019). Moreover, a recent study showed that neurological disease–associated mutations of EXOSC3 indeed caused pre-rRNA processing defects when introduced into Rrp40, the EXOSC3 homologue in budding yeast (Gillespie et al, 2017). Likewise, HITS-CLIP (high-throughput sequencing of RNA isolated by cross-linking immunoprecipitation) identified (pre-)rRNAs as major targets of the human exosome (Belair et al, 2019). In line with these publications, one of the main findings of our RNAseq analysis was that one-third of all increased protein-coding transcripts relate to ribosome assembly and rRNA processing; furthermore, we observed an increase in the levels of scaRNAs and snoRNAs, which are important for pre-rRNA processing and modification (Watkins & Bohnsack, 2012). Up-regulation of RNA and protein factors important for key steps during ribosome synthesis might, therefore, be a compensation mechanism for a defect in rRNA processing caused by exosome deficiency.

Defects in the assembly of the ribosome cause ~20 distinct genetic diseases known as ribosomopathies (reviewed in Aubert et al [2018]). Like the exosome, the ribosome is highly abundant and ubiquitously expressed, yet ribosomopathies are characterised by cell- and tissue-specific defects due to improper cellular development and cell death, such as anaemia and skeletal defects (e.g., Diamond–Blackfan anaemia), craniofacial defects (Treacher Collins syndrome) or microcephaly (Bowen–Conradi syndrome). Moreover, patients with ribosomopathies are often pre-disposed to cancer because of misregulation of p53 and downstream signalling pathways, which affect the cell cycle and cellular survival. RNAi-mediated knockdown of EXOSC8 or EXOSC9 in HCT116 cells resulted in stabilisation of the p53 protein, which, in the case of EXOSC9, was demonstrated to be alleviated by co-depletion of the 5S RNP component RPL5. This indicated that the impact of core exosome depletion on p53-dependent signalling is not just due to increased levels or stability of the p53 mRNA, as seen in our mutant zebrafish and patient muscle samples, but also linked to the essential function of the exosome in the processing of the rRNA. Although future studies will dissect the multiple ways by which p53 levels may be affected in the absence of a functional core exosome, and to what extent this is carried out at the mRNA and/or protein level, our data strongly suggest that diseases characterised by exosome deficiencies should also be considered as ribosomopathies. Furthermore, EXOSC9 depletion also caused a G2/M cell cycle arrest reminiscent of a nucleolar stress response, which may slow down cell proliferation and eventually lead to apoptosis, consistent with the phenotype seen in the zebrafish model.

Embryos with homozygous *exosc8* and *exosc9* mutations develop a small head, and have less brain tissue, and a smaller cerebellum. Acridine orange staining showed that there was increased apoptosis in the mutant embryos, supported also by RNAseq and qRT-PCR findings, suggesting that elevated rates of cell death in the brain play a key role in pathogenesis of exosome mutations. We tried to rescue the increased cell death in the brain of *exosc8* and *exosc9* mutant fish by injecting an anti-p53 morpholino oligonucleotide at the one-cell stage and monitored the head size at 5 dpf;

however, it turned out that injecting the p53 morpholino into wild-type fish led to a decreased head size itself, making the phenotype rescue impossible (data not shown).

We also observed that *exosc8* and *exosc9* homozygous mutant embryos develop craniofacial abnormalities. In patients with exosome subunit mutations, craniofacial abnormalities have only been reported in patients with *EXOSC2* mutations (Di Donato et al, 2016). However, a very recent publication also reported *EXOSC9* patients with facial dysmorphic features (Bizzari et al, 2019), indicating that the disease spectrum could be overlapping. As the number of reported patients with exosome mutations is not very high, the full spectrum of disease symptoms might not be known yet.

Craniofacial defects are common in ribosomopathies: both Treacher Collins syndrome and Diamond–Blackfan anaemia present with craniofacial hypoplasia (Ross & Zarbalis, 2014). Various studies have demonstrated that mutations in nucleolar proteins involved in rRNA synthesis or ribosomal subunit assembly affect ribosome biogenesis and activate a p53 response through a nucleolus-mediated stress (Ross & Zarbalis, 2014). Cranial neural crest cells, which form bone, cartilage, and connective tissue in the head, are especially sensitive to nucleolar stress (Ross & Zarbalis, 2014; Calo et al, 2018). Ribosome assembly defects might leave neural crest cells unable to produce essential new proteins required for their developmental programme or migration (Calo et al, 2018). For example, zebrafish with mutations in 18S rRNA factor 1 (esf1) have severe pharyngeal cartilage loss due to deficits in cranial neural crest cell development (Chen et al, 2018). Esf1 mutants also show increased apoptosis and p53 signalling. Similarly, mutations in zebrafish Wdr43, another ribosome biogenesis factor, also result in cartilage loss due to neural crest cell defects (Zhao et al, 2014). The cartilage defects seen in *exosc8* and *exosc9* mutants are, therefore, likely due to neural crest cell development defects. Both esf1 and Wdr43 transcripts are among the increased ribosome biogenesis factors in our mutant zebrafish, suggesting that up-regulation of these proteins may contribute to the previously suggested mechanism to compensate for defects in rRNA processing caused by exosome deficiency. Interestingly, mutations in *pescadillo*, another differentially expressed transcript in our mutant zebrafish, disrupt oligodendrocyte formation and cause a loss of myelination in zebrafish (Simmons & Appel, 2012). Pescadillo is required for nucleolar assembly and ribosome biogenesis (Lerch-Gaggl et al, 2002); its loss affects cell cycle progression and migration of oligodendrocytes in zebrafish embryos. We have previously shown that *exosc8* morphant zebrafish and patients with *EXOSC8* mutations had myelination defects (Boczonadi et al, 2014), which might also relate to a perturbation of ribosome biogenesis.

Transcripts encoding ribosome biogenesis factors were not differentially expressed in iNPCs derived from patient fibroblasts, suggesting that, in contrast to our mutant zebrafish, iNPCs do not alter gene expression to compensate for defects in rRNA processing caused by exosome deficiency. We found, however, a dysregulation of scaRNAs and snoRNAs in iNPCs, which are essential for co- and post-transcriptional 2′-O-methylation of rRNA, snRNA, and tRNA (Meier, 2017). Notably, some snoRNAs are essential for pre-rRNA processing (Watkins & Bohnsack, 2012).

Dysregulation of scaRNAs and snoRNAs in the iNPCs may, therefore, also directly and/or indirectly affect rRNA processing and ribosome assembly. Mutations in Essential for Mitotic Growth 1 (EMG1), a ribosome biogenesis factor involved in the maturation of the 18S rRNA, cause the ribosomopathy Bowen–Conradi syndrome (Armistead et al, 2009, 2014). EMG1 mutations only caused cell cycle arrest in fast proliferating lymphoblasts, but not in slowly proliferating fibroblasts (Armistead et al, 2014). This publication indicates that the extent of manifestation of a ribosome biogenesis defect, and, consequently, its downstream effects on gene expression and cellular signalling, greatly depends on the proliferation rate of the cells.

In addition, the residual amount of functional exosome complex still present in our patient cells might be higher than in the siRNA-induced depletion in the studies mentioned above, thus explaining the less striking changes in transcript levels detected in our patient cells. A recent publication has shown that the exosome is down-regulated when ES cells start to differentiate and up-regulated when fibroblasts are reprogrammed to induced pluripotent stem cells (Belair et al, 2019). As the experimental setup for the conversion of fibroblasts to iNPCs relies on similar mechanisms as the reprogramming to induced pluripotent stem cells, presumably the exosome is also up-regulated during the conversion performed by us. Residual amount of exosome might be enough to prevent a severe impairment of ribosome biogenesis in iNPCs. Furthermore, cellular environment and growth factor signalling are different in cell culture compared with the in vivo context; cellular requirements change dramatically once cells differentiate or migrate during development. The discrepancies between our results in cultured patient cells and zebrafish models for exosome-related diseases emphasize the importance of animal models to study these complex diseases.

Zebrafish models have also been generated for other PCH subtypes, such as PCH caused by mutations in *CLP1*, *TOE1*, and *tsen54*. *Clp1* zebrafish mutants had an abnormal head shape, impaired movement abilities, and died before 5 dpf (Schaffer et al, 2014). Morpholino-induced knockdowns of *toe1* and *tsen54* in zebrafish led to a phenotype with small heads, small eyes, curved tails, and structural defects of the developing midbrain, cerebellum, and hindbrain (Kasher et al, 2011; Lardelli et al, 2017). Furthermore, *Clp1* mutants and *tsen54* morphants also had increased cell loss in the forebrain and hindbrain, leading to decreased levels of the midbrain marker *otx2* from 48 hpf onwards (Kasher et al, 2011; Schaffer et al, 2014), as observed in the transcript analysis in our exosome mutant fish. Although CLP1, TOE1, and the TSEN complex are not linked to ribosome biogenesis, the similar phenotype in patients and animal model points towards a converging disease mechanism in PCH.

In summary, our results provide new insights into the mechanism behind pathogenesis in exosome-related diseases and further establish the de-regulation of RNA pools and ribosome biogenesis as a common disease pathway in PCH and motor neuron disorders. Perturbation of RNA levels and ribosome biogenesis leads to p53 up-regulation, cell cycle arrest, and increased cell death in the brain. Further research, however, is needed to understand why certain cell types in the brain are more vulnerable than others.

# Materials and Methods

### Zebrafish strains and husbandry

All zebrafish used in this study were the *golden* strain. All procedures carried out on zebrafish were approved by the Home Office UK. Zebrafish embryos were collected and raised at 28.5°C in E3 medium using established procedures and staged in hpf or dpf (Kimmel et al, 1995).

### single (s)gRNA synthesis

Crisprscan (Moreno-Mateos et al, 2015) was used to identify a target site in exon 2 of *exosc8* and exon 3 of *exosc9* in zebrafish. sgRNA was produced as described elsewhere (Varshney et al, 2015). An oligonucleotide with a T7 promoter, target sequence, and a complementary sequence was annealed to a bottom strand "ultramer" oligo in a thermocycler and extension of the oligonucleotides was catalysed by DNA polymerase (Mytaq) to form the template oligonucleotide for sgRNA synthesis (Table S7). The sgRNA template oligonucleotide was purified using a QIAGEN PCR purification kit. sgRNA was synthesised from the sgRNA oligonucleotide template using the MEGAshortscript T7 Kit (Ambion) and purified with the mirVana RNA isolation kit (Thermo Fisher Scientific) and stored at –80°C until required for injection.

### Injection of sgRNA/Cas9

The sgRNA was diluted to 300 ng/µl with 2 µM Cas9 protein (New England Biolabs), 2M KCl and 0.05% phenol red, and heated to 37°C for 5 min. Freshly laid embryos were injected up until the two-cell stage with 1 nl of gRNA.

### Establishing the *exosc8* and *exosc9* zebrafish lines

Injected F0 mosaic founder fish were raised to adulthood with some clutchmates euthanized to confirm mutagenesis. To confirm mutagenesis genomic DNA was extracted from pools of 5–10 embryos per clutch at 5 dpf using the "hotSHOT" technique (Meeker et al, 2007). Embryos were lysed in 20 µl of 50 mM NaOH (Sigma-Aldrich) for 30 min at 95°C, periodically vortexing. The lysate was then neutralized with 20 µl 100 mM Tris–HCl (Sigma-Aldrich). PCR was then performed on the lysates using the primers listed in the supplementary data. The PCR products were ligated into the pGEM-T easy vector (Promega) following the manufacturer's instructions. JM109 High Efficiency Competent Cells (Promega) were transformed with the ligated plasmid and plated on ampicillin containing LB agar plates. Plates were incubated at 37°C overnight. Colony PCR was then performed on plasmids using standard pUC/M13 primers (Eurofins, supplementary data). PCR products were sequenced to confirm that random indels had occurred in the target regions.

Once F0 mosaic zebrafish were of breeding age they were outcrossed with wild type *golden* zebrafish to produce the F1 generation of zebrafish. These zebrafish were raised to adulthood and a biopsy of the tail fin was collected under anaesthesia with tricaine (140–164 mg/litre). Genomic DNA was extracted using the

"hotShot" technique (Meeker et al, 2007) and PCR and sequencing was performed to detect F1 heterozygotes (Fig S1). F1 heterozygous zebrafish were outcrossed with wild-type golden zebrafish to produce F2 zebrafish, which were raised and then genotyped when adults to identify the heterozygotes. These F2 zebrafish were incrossed to produce embryos used in subsequent studies. Live embryos were imaged using a Leica MZ16 microscope.

## Whole mount immunofluorescence in zebrafish embryos

Zebrafish used for immunofluorescence were treated with 0.003% 1-phenyl-2-thiourea (Sigma-Aldrich) from 24 hpf to prevent pigment formation and aid imaging. Zebrafish embryos were terminally anaesthetized at 5 dpf using tricaine and then fixed overnight at 4°C in 4% paraformaldehyde in PBS. Embryos were washed in PBS plus 0.1% Tween 20 (PBT; Sigma-Aldrich) and then permeabilized with acetone for 30 min at −20°C. Embryos were also permeabilized with 0.25% trypsin for 90 min. Embryos were then blocked in blocking solution (5% horse serum in PBT) for 1 h. Embryos were incubated in the primary antibodies diluted in the blocking solution (SV2, 1:200; Developmental Studies Hybridoma Bank; HuC, 1:200; Abcam; pvalb7 1:1,000) overnight at 4°C. After washing with PBT, embryos were incubated with secondary antibodies (anti-mouse Alexa Fluor 488, anti-rabbit Alexa Fluor 594, or anti-rabbit Alexa Fluor 488; Invitrogen) diluted in blocking solution for 1 h. Phalloidin and α-bungarotoxin (Thermo Fisher Scientific) were conjugated to Alexa Fluor 594 and did not require any secondary antibodies. Immunofluorescent images were captured using a Zeiss Axio Imager with Zen software. The Parvalbumin7 antibody was a kind gift of Prof. Masahiko Hibi, Nagoya University, Japan.

## Alcian blue staining

Zebrafish used for Alcian blue staining were treated with 0.003% 1-phenyl-2-thiourea (Sigma-Aldrich) from 24 hpf to prevent pigment formation and aid imaging. Zebrafish embryos were terminally anaesthetized at 5 dpf using tricaine (200–300 mg/ml; Sigma-Aldrich). Embryos were then fixed overnight at 4°C in 4% paraformaldehyde in PBS. Embryos were washed in PBS and then dehydrated in 50% ethanol then stained overnight with Alcian blue (0.02% Alcian blue [Sigma-Aldrich], 60 mM $MgCl_2$ in 70% ethanol). Zebrafish were washed in PBS and then left for the tissue to clear overnight in 0.25% KOH. Zebrafish were washed in PBS and then imaged using a Leica stereo microscope. Image analysis was performed using ImageJ (Schneider et al, 2012b).

## Acridine orange staining

At 48 hpf zebrafish embryos were placed in 5 µl/ml acridine orange (Sigma-Aldrich) for 30 min and then washed in aquarium water. Embryos were imaged using a Zeiss Axio Imager with Zen software.

## Cell culture

Primary skin fibroblasts of patients carrying the homozygous p.(Leu14Pro) in EXOSC9, the homozygous p.(Ala2Val) in EXOSC8, the

homozygous p.(Gly31Ala) in EXOSC3, and control fibroblast lines were grown in high-glucose DMEM (Thermo Fisher Scientific) supplemented with 10% FBS as previously described in Burns et al (2018). The patient with the homozygous EXOSC8 mutation (patient from pedigree 3 in Boczonadi et al [2014]) presented with spinal muscular atrophy and cerebellar and corpus callosum hypoplasia. The patient with EXOSC3 mutations suffered from a severe form of PCH type 1 that included severe hypotonia, weakness, absent motor and mental development, and signs of neuronal loss in spinal anterior horns, cerebellum, and parts of the midbrain (Schwabova et al, 2013). The patient carrying an EXOSC9 mutation (patient 1 in Burns et al [2018]) presented with muscle weakness, hypotonia, axonal motor neuronopathy, and mild cerebellar atrophy; in general, her clinical symptoms were milder than those of patients with EXOSC8 and EXOSC3 mutations. The control fibroblasts were obtained from genetically unrelated, healthy adult donors (two female and one male) aged between 20 and 40 yr.

## Direct conversion of skin fibroblasts to iNPCs

Primary fibroblasts of the EXOSC3, the EXOSC8, the EXOSC9 patient and four control fibroblasts were converted to neuronal progenitor cells using the method published by Meyer et al (2014), (Fig S5A–F). Successful conversion was confirmed by gene expression analysis for neuronal and fibroblast markers where an up-regulation in the expression of the neural stem cell markers Sox2, Sox1, nestin, and Pax6 and a down-regulation of the fibroblast markers Col1a1 and Col3a1 was observed (Fig S5G). One colony was picked from each iNPC line for further expansion and RNAseq analysis.

## RNAi-mediated down-regulation of EXOSC8 or EXOSC9

HCT116 cells were cultured in DMEM (Sigma-Aldrich), supplemented with 10% FBS (Sigma-Aldrich), and 1% penicillin/streptomycin (Sigma-Aldrich). RNAi-mediated depletion in HCT116 cells was performed by reverse transfection of siRNA duplexes targeting the mRNA sequence of EXOSC8, EXOSC9, or RPL5 using Lipofectamine RNAiMAX transfection reagent (Life Technologies). An siRNA targeting firefly luciferase mRNA (GL2) was used as a negative control. Approximately 80,000 cells were seeded in 24-well plates (Sarstedt) and incubated with the siRNA(s) for ~60 h before being harvested for protein analysis. The following siRNA oligos were used: control (GL2): 5′-CGUACGCGGAAUACUUCGATT (Elbashir et al, 2002); EXOSC8: 5′-CGACUACGAUGGAAACAUUTT, EXOSC9: 5′-GCUACUAAAAGAUCAA-GUUTT (both from Tafforeau et al [2013]); RPL5: 5′-GGUUGGCCU-GACAAAUUAUTT. All siRNA oligos were obtained from Eurofins MWG.

## SDS–PAGE and immunoblotting

HCT116 cells grown in 24-well plates were pelleted by centrifugation and resuspended in ~50 µl of 2× protein loading dye (75 mM Tris–HCl, pH 6.8, 1.25 mM EDTA, 20% glycerol, 2.5% SDS, 0.125% bromophenol blue, and 50 mM DTT). 10 µl of each sample was loaded on a 1.5 mm denaturing SDS 13% polyacrylamide gel (Bio-Rad), transferred to a nitrocellulose membrane (Protran; GE Healthcare) and probed with monoclonal primary antibodies recognising EXOSC8, EXOSC9, p53, karyopherin β1, or GAPDH, diluted

in PBS-Triton X-100 (0.1%, vol/vol) containing 2% non-fat dried milk (wt/vol) (Marvel). An IRDye-labelled secondary antibody (LI-COR) was used, and membranes were visualised using the Odyssey LI-COR system (LI-COR). Quantification was performed using Image-Quant software (GE Healthcare) and quantified protein levels were normalised to levels of the loading control, karyopherin. The following antibodies were used: $\alpha$-EXOSC8 (mouse, (H-8) sc-393027; Santa Cruz), dilution 1:1,000; $\alpha$-EXOSC9 (mouse, (D-6) sc-271815; Santa Cruz), dilution 1:1,000; $\alpha$-p53 (mouse, (DO-1) sc-126; Santa Cruz), dilution 1:500; $\alpha$-karyopherin $\beta 1$ (mouse, (H-7) sc-137016; Santa Cruz), dilution 1:2,000; $\alpha$-GAPDH (mouse, (0411) sc-47724; Santa Cruz), dilution 1:10,000; and secondary antibody Donkey $\alpha$-Mouse 800CW LI-COR (926-32212), dilution 1:10,000.

### PI staining and flow cytometry

After 60-h siRNA-mediated depletion of EXOSC9 or overnight (18 h) actinomycin D (ActD, 4 nM) treatment, HCT116 cells were pelleted by centrifugation, washed in PBS containing 1% FBS, and fixed in ice-cold 70% ethanol. Fixed cells were resuspended in RNase A (Sigma-Aldrich) diluted in water (100 $\mu$g/ml). PI (Invitrogen), diluted in PBS, was added to a final concentration of 50 $\mu$g/ml. Cells, in a total volume of 250 $\mu$l, were incubated in the dark for 20 min at 4°C before analysis by flow cytometry using the FACS Canto II machine and analysed using FACSDIVA software (BD Biosciences).

### RNA extraction

RNA from cultured cells or zebrafish was extracted with Trizol reagent (Thermo Fisher Scientific) following the manufacturer's instructions. RNA extraction was followed by DNAse treatment with the DNA-free DNA Removal Kit (Ambion).

For downstream RNAseq analysis, RNA from cells and zebrafish embryos was extracted using the mirVana miRNA isolation kit (Thermo Fisher Scientific) following the manufacturer's instructions. Embryos were euthanised at 5 dpf in tricaine before RNA extraction. Pools of ~30 embryos per experimental group were collected. RNA extraction for RNAseq analysis was performed in triplicate for each sample (i.e., three cell pellets or three zebrafish embryo pools). Three biological replicates for each mutant and for the controls (3 batches of 30 pooled embryos each) were used for the RNA analysis.

### RNA sequencing

RNAseq libraries were prepared with Illumina TruSeq Stranded polyA–enriched RNA with Ribo-Zero Human kit and sequenced as previously described (Burns et al, 2018). *Exosc3* morphant zebrafish and control zebrafish RNAseq reads were downloaded from European Nucleotide Archive Study: PRJNA470927 (Francois-Moutal et al, 2018). Raw sequence reads were trimmed to 50 bp with FastX-toolkit v.0.0.14 (http://hannonlab.cshl.edu/fastx_toolkit/index.html) and aligned to complete zebrafish (danRer11) or human (hg38) reference genomes, using the STAR aligner v.2.5.3a two-pass protocol (Dobin et al, 2013). Differential expression analysis was performed with HTSeq v.0.9.1 (Anders et al, 2015) and DESeq2 v.1.12.4 (Love et al, 2014). Gene set over-representation analysis was performed via http://cpdb.molgen.mpg.de/. Bedfiles of

non-coding RNA types were prepared from gene coordinates available at Ensembl Biomart except for cytosolic tRNAs, where sequences and coordinates were obtained from GtRNAdb (http://gtrnadb.ucsc.edu/). BEDTools v.2.26.0 was used to obtain per gene strand–specific counts of non-coding RNAs. RNA sequencing files were deposited to National Center for Biotechnology Information Gene Expression Omnibus under accession number GSE151452.

RNAseq data from muscle samples of two *EXOSC9* patients has been published previously (Burns et al, 2018).

### qRT-PCR

To confirm the RNAseq results, quantitative real-time PCR (qRT-PCR) was carried out using a CFX96 RealTime qPCR machine (Bio-Rad). The SuperScript III First-Strand Synthesis System (Thermo Fisher Scientific) was used to produce cDNA from 1 $\mu$g of the isolated RNA following the manufacturer's instructions. qRT-PCR was performed using the primers listed in the supplementary data using iTaq Universal SYBR Green Supermix (Bio-Rad). Each reaction was carried out in triplicate and elongation factor 1 alpha (*ef1α*, zebrafish) or GAPDH (human cells) was selected as an internal control to normalize the RNA levels. The $2^{-\Delta\Delta Ct}$ method was used to quantify gene expression. Three biological replicates for each mutant and for the controls (3 batches of 30 pooled embryos each) were used for the RNA analysis.

## Supplementary Information

## Acknowledgements

This work was supported by the Medical Research Council (UK) (MR/N025431/1 to R Horvath), the Wellcome Investigator Fund (109915/Z/15/Z to R Horvath), the Newton Fund (UK/Turkey, MR/N027302/1 to R Horvath), the Wellcome Centre for Mitochondrial Research (203105/Z/16/Z to R Horvath), the Royal Society (UK) (UF150691 to C Schneider), and the Biotechnology and Biological Sciences Research Council/Medical Research Council (UK) (BB/R00143X/1 to C Schneider). The authors wish to thank Dr Denisa Hathazi for helping to design the graphical abstract and Nandor Baranyi, Alba Vilella, Daniel Cox, and the staff of the zebrafish facility at Newcastle University for their help with zebrafish husbandry and genotyping.

### Author Contributions

JS Müller: conceptualization, formal analysis, supervision, funding acquisition, investigation, methodology, and writing—original draft, review, and editing.
DT Burns: conceptualization, investigation, methodology, and writing—original draft.
H Griffin: conceptualization, data curation, formal analysis, investigation, and writing—original draft.
GR Wells: investigation and methodology.
RA Zendah: investigation and methodology.

B Munro: conceptualization, supervision, funding acquisition, investigation, methodology, project administration, and writing—original draft, review, and editing.

C Schneider: conceptualization, formal analysis, supervision, funding acquisition, investigation, methodology, project administration, and writing—original draft, review, and editing.

R Horvath: conceptualization, formal analysis, supervision, funding acquisition, investigation, project administration, and writing—original draft, review, and editing.

## Conflict of Interest Statement

The authors declare that they have no conflict of interest.

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
