## [Reviewer comments · Life Science Alliance]

Life Science Alliance

RNA exosome mutations in pontocerebellar hypoplasia alter ribosome biogenesis and p53 levels

Juliane Mueller, David Burns, Helen Griffin, Graeme Wells, Romance Zendah, Benjamin Munro, Claudia Schneider, and Rita Horvath

DOI: <https://doi.org/10.26508/lsa.202000678>

Corresponding author(s): Rita Horvath, University of Cambridge and Claudia Schneider, Newcastle University

Review Timeline:

Submission Date:	2020-02-17
Editorial Decision:	2020-03-30
Revision Received:	2020-04-24
Editorial Decision:	2020-05-22
Revision Received:	2020-06-01
Accepted:	2020-06-03

Transaction Report:

March 30, 2020

Re: Life Science Alliance manuscript #LSA-2020-00678-T

Prof. Rita Horvath
University of Cambridge
Department of Clinical Neurosciences
University Neurology Unit, Level 5 'A' Block, Box 165
Cambridge Biomedical Campus
Cambridge CB2 0QQ
United Kingdom

Dear Dr. Horvath,

Thank you for submitting your manuscript entitled "RNA exosome mutations in pontocerebellar hypoplasia alter ribosome biogenesis and p53 levels" to Life Science Alliance. The manuscript was assessed by expert reviewers, whose comments are appended to this letter.

As you will see, the reviewers appreciate your data, but think that some of them are too preliminary at this stage or contradicting effects one would expect upon loss of exosome function. They further note that the different parts of the manuscript are ill-connected. All reviewers provide constructive input on how to address their concerns, and we would thus like to invite you to submit a revised version to us. Importantly, quantifications and statistical analyses need to get provided throughout and the number of experiments should get stated (all reviewers). The human cell line analyses need inclusion of p53 transcript levels to better link the results to the previous zebrafish-based ones (rev#1), and the data in Figure 4 are only preliminary and should get replicated (rev#2). Some re-analysis of the existing data and changes in their representation are needed as well.

We are aware that many laboratories cannot function fully during the current COVID-19/SARS-CoV-2 pandemic and therefore encourage you to take the time necessary to revise the manuscript to the extent requested above, as we strive for robustness for the work published in Life Science Alliance. We will extend our 'scoping protection policy' to the full revision period required. If you do see another paper with related content published elsewhere, nonetheless contact me immediately so that we can discuss the best way to proceed. Please also get in touch in case you would like to discuss how to best address the above-mentioned issues in light of the current situation.

Please note that papers are generally considered through only one revision cycle, so strong support from the referees on the revised version is needed for acceptance.

Thank you for this interesting contribution to Life Science Alliance. We are looking forward to receiving your revised manuscript.

Sincerely,

B. MANUSCRIPT ORGANIZATION AND FORMATTING:

*****IMPORTANT:** It is Life Science Alliance policy that if requested, original data images must be made available. Failure to provide original images upon request will result in unavoidable delays in

publication. Please ensure that you have access to all original microscopy and blot data images before submitting your revision.***

Reviewer #1 (Comments to the Authors (Required)):

In the manuscript by Müller et al. entitled "RNA exosome mutations in models of pontocerebellar hypoplasia alter ribosome biogenesis and p53 levels leading to apoptosis", the authors employ a series of complementary systems and approaches to provide insight into why mutations in genes encoding structural subunits of the RNA exosome cause a spectrum of human diseases. Studies to define the mechanisms that underlie the clinical manifestations in diseases caused by mutations in RNA exosome genes are very important. The types of studies presented here will be critical to provide such insight.

The authors focus primarily on new zebrafish models, but they complement this approach with knockdown in cultured cells as well as induced neuronal progenitor cells (iNPCs) produced from patient fibroblasts. While the authors detect changes in a number of coding and noncoding transcripts, at least one major conclusion highlighted in the title does not seem to hold true in the iNPCs. Comparing different systems is challenging, but these cells are the only that truly model the disease, which is caused by mutant rather than absent proteins so this raises some concerns about a major conclusion of the work.

The RNA exosome both decays and processes RNAs. The direct targets of the RNA exosome are likely to show an increase in response to loss of RNA exosome function. The authors fail to appreciate this mechanistic point in some of their analysis. In fact, the results of the RNA-seq experiment are rather surprising because the authors detected more transcripts that were decreased in the RNA exosome mutant fish than transcripts increased. The transcripts that increase are most likely to be the direct targets of the RNA exosome. Indeed, among the most critical roles of the RNA exosome is certainly rRNA processing to produce the mature rRNA required to assemble functional ribosomes. The authors refer to this point, but do not cite the one study, which demonstrates that mutations that model those found in EXOSC3 cause defects in ribosomal RNA processing (Gillespie A, Gabunilas J, Jen JC, Chanfreau GF. RNA. 2017 Apr;23(4):466-472. Mutations of EXOSC3/Rrp40p associated with neurological diseases impact ribosomal RNA processing functions of the exosome in *S. cerevisiae*. doi: 10.1261/rna.060004.116. Epub 2017 Jan 4. PMID: 28053271). This point supports the authors' supposition and logical suggestion that mutations in the RNA exosome could fall into the category of ribosomopathies.

Specific Comments:

In the Introduction and some previous work, the authors state that "In the cytoplasm, the exosome degrades mRNAs that contain AU-rich elements (AREs) and RNAs that have evaded degradation in the nucleus" and reference a review article.

The authors have generated homozygous crispant/knockout zebrafish to model loss of EXOSC9 and EXOSC8. The authors show sequencing indicating that they have created what should be early nonsense mutations in both genes. They cross these heterozygous fish to produce homozygous fish that should have these non-functional alleles of RNA exosome subunits. The authors analyze these embryos at 5 dpf for most of the subsequent studies. The authors do not provide any information about the viability of these embryos. This point is an important one as no

studies have used genetic mutants (rather than depletion) to convincingly show whether the RNA exosome is essential in multicellular organisms. The authors have the opportunity to address this important point.

With regard to the phenotypes analyzed here, the authors state that 25% of the clutch appear to have smaller head and eyes. The data presented (Figure Legend and/or figure) should include the n values of how many embryos were analyzed to produce the data presented in Figure 1C,D,E.

The authors provide genomic PCR results to confirm the homozygous alleles. While no antibodies are available to confirm the loss of the EXOSC proteins in these fish, the authors need to perform qRT-PCR to evaluate levels of the exosc8 and exosc9 transcripts. An endpoint RT-PCR data presented in the supplemental data shows a decrease in steady-state RNA levels in for exosc9, but the authors state that they detect no difference in the levels of the exosc8 transcript.

The authors then went on to use these fish to perform RNA-seq analysis. The authors need to clearly state how many independent samples were analyzed for this RNA-seq, including wildtype or het controls. The methods state that they employed ~30 embryos for each sample- does this constitute a single ensemble biological replicate? At least two replicates should be performed to ensure that differences detected represent real changes rather than variability across the samples.

Typically for an RNA-Seq analysis, a volcano plot is employed to illustrate the overall changes in transcripts. Given that the RNA exosome is an RNA decay/processing machine, the most direct effects would likely result in an increase in the steady-state levels (via stabilization) of those transcripts. As mentioned above, the finding that more transcripts decrease in both zebrafish models rather than increase raises some concern about whether these changes might reflect indirect effects. The authors need to address this point and speculate about why they see more transcripts decrease than increase. Figure 2E seems like it could be omitted or moved to the supplemental data.

The authors focus on a modest increase in the level of p53 pathways transcripts. They only validate the result for the p53 transcript in Figure 2G. The data presented in this figure needs to include statistical analysis.

Figure 5 employs knockdown in cultured human cells to explore the impact of loss of EXOSC9 on p53 levels. The authors' analysis of the zebrafish transcriptome that led them to investigate the increase in p53 levels revealed an increase in the p53 transcript that they cite as a 2.44 fold increase (Figure 2F). This increase is validated by qRT-PCR in Figure 2G (albeit whether this increase is statistically significant is not indicated in this figure). In Figure 5A, the authors show that knockdown of EXOSC9 in HCT116 colon cancer cells causes a 3-fold increase in levels of p53 protein. They then state that this could "reflect increased stability and/or higher translation rate of the p53 mRNA in HCT116 cells in the absence of an intact core exosome." The authors need to perform qRT-PCR to examine p53 transcript levels in these knockdown cells. In the zebrafish studies, reagent limitations mean that the authors cannot correlate protein and transcript levels, but here they can and should to provide mechanistic insight into the changes in p53 levels. They could readily examine p53 transcript levels of protein levels under the experimental conditions employed to ascertain the regulatory mechanism. Despite the result with concomitant knockdown of RPL5- there are multiple other interpretations. The most logical one is- the p53 transcript increases about 3-fold (likely due to increased stability if directly regulated by the RNA exosome, an RNA decay machine) and thus the protein levels also increase 3-fold. An effect on the stability of the p53 protein seems convoluted and perhaps an indirect effects of loss of RNA exosome activity.

To reorganize this manuscript without the need for additional experiments, the authors could present Figure 1, Figure 3, Figure 4 (suggesting that apoptosis may occur in small eyes/heads does not require rationale of changes in p53), Figure 7. These studies all characterize the morphology of the zebrafish embryos. The authors could then present the RNA-Seq data in Figure 2 (modifying the presentation to include volcano plots). The Discussion could then mention the logical transcript changes from the RNA-seq that correlate with the morphological changes detected. This approach would eliminate some of the work that absolutely requires additional data to support the conclusions and would also eliminate the iNPCs, which are potentially very exciting, but do not really fit with the narrative here.

Minor points:

In the Introduction and some previous work, the authors state that "In the cytoplasm, the exosome degrades mRNAs that contain AU-rich elements (AREs) and RNAs that have evaded degradation in the nucleus" and cite reference 16 (van Hoof A, Lennertz P, and Parker R. Yeast exosome mutants accumulate 3'-extended polyadenylated forms of U4 small nuclear RNA and small nucleolar RNAs.

Molecular and cellular biology. 2000;20(2):441-52).

This reference does not seem to provide any suggestion that targets of the RNA exosome are ARE-containing transcripts and this model is not typically embraced by more recent studies of the RNA exosome.

Another study often (and logically) cited to support this model is another somewhat dated study:

Devi Mukherjee 1, Min Gao, J Patrick O'Connor, Reinout Raijmakers, Ger Pruijn, Carol S Lutz, Jeffrey Wilusz. EMBO J, 21 (1-2), 165-74 2002 Jan 15 The Mammalian Exosome Mediates the Efficient Degradation of mRNAs That Contain AU-rich Elements

The results reported here likely reflect the fact that ARE-containing transcripts are often short-lived. Hence, impairing RNA exosome function has an outsized effect in stabilizing transcripts that have a shorter half-life as compared to more stable transcripts. There is little evidence that the RNA exosome complex has specificity for any specific RNA sequences.

The authors refer to "upregulated" genes, a term that has come from the many studies that explore transcriptional changes through RNA-seq analysis. Better terminology here might be to refer to an increase in steady-state levels or just use increase. The term upregulated implies an increase in response to an action- as the RNA exosome decays RNA, the more likely scenario is that loss of RNA exosome function leads to the stabilization of transcripts- rather than upregulation per se. A minor point- but an important one.

Reviewer #2 (Comments to the Authors (Required)):

The exosome is a multipart protein RNA exonuclease complex that functions as an essential regulator of cellular RNA processing and metabolism. To date, multiple genetic variants in core

exosome proteins or exosome cofactors have been observed in patients diagnosed with neurodegenerative diseases including pontocerebellar hypoplasia (PCH) and motor neuropathy. Previous work in the field has demonstrated that in patient cells these mutations (1) reduce protein levels, stoichiometrically restricting formation of functional exosome complexes, and (2) result in perturbations of the transcriptome consistent with patient presentations. Additionally, morpholino depletion of these factors in zebrafish has recapitulated muscle weakness and PCH-like phenotypes. However, the full transcriptomic and developmental consequences following complete loss of EXOSC8 or EXOSC9 have yet to be modeled.

In this work, Müller et al. present confirmatory evidence that EXOSC8 and EXOSC9 are critical for normal craniofacial development and transcriptome maintenance by analyzing CRISPR knockout zebrafish. Furthermore, the authors demonstrate that EXOSC9 depletion in human cells stabilizes TP53 in an RPL5-dependent manner, and causes G2/M arrest reminiscent of the nucleolar stress response. Consistent with these observations, transcripts associated with ribosome biogenesis and assembly were significantly upregulated in null EXOSC8 and EXOSC9 zebrafish, possibly as a compensatory mechanism for impaired ribosome biogenesis. Lastly, the authors have shown preliminary data that craniofacial-specific apoptosis, a hallmark of a subset of ribosomopathies, is occurring in the mutant fish. Taken together, Müller et al. argue that diseases like PCH and motor neuropathy arising from mutations in exosome proteins or cofactors can be viewed as ribosomopathies, which is consistent with the importance of the exosome in pre-ribosomal RNA (pre-rRNA) processing.

This manuscript presents both confirmatory and novel evidences for the role of the exosome in development and ribosome biogenesis. In particular, the assertion that diseases caused by exosome mutations are ribosomopathies is valuable and expands the growing list of diseases in which ribosome biogenesis is implicated. However, this claim is mostly predicated on evidence in Figure 2A-B, Figure 4, and Figure 5, which only begin to implicate exosome diseases as ribosomopathies. In particular, the data in Figure 4 is only preliminary and should be replicated before inclusion in published work (see comment 2), and the effect of EXOSC8 depletion on TP53 stabilization is not examined (see comment 3). Additional experiments analyzing defects in pre-rRNA transcription and processing would also strongly support this conclusion, although those could be tabled as future directions. Even more importantly, this manuscript would greatly benefit from an overhaul of its quantitative and statistical data treatments (see comment 1). There are inconsistent explanations for what measures of centrality and error are being graphed, what the sample size for experiments is, and what statistical tests are being performed. These issues should be addressed by the authors before publication.

Major comments

1. Figure 1, 2, 3, 5, 7, S2: Please make clear which measures of centrality and error are being represented graphically, and which (appropriate) statistical tests are being conducted on data in these figures. Figure 5 is the only place any mention of statistical procedures is mentioned. Additionally, t-tests are not appropriate when testing differences between more than two means as in 1C-E, 3B-C, 5A/C, 7B-D, and S2B; please use an appropriate test like one-way ANOVA followed by a post-hoc test.

2. Figure 4: The claim that KO fish exhibit increased apoptosis would be strengthened by including quantification of the acridine orange staining across multiple fish. As-is, $n = 1$ for each condition is preliminary and is not adequate to strongly make this claim, and cannot reveal if EXOSC8 or

EXOSC9 is more important for this hallmark phenotype of ribosomopathies. The observation of upregulated craniofacial-specific apoptosis, rather than simple failure of those cells to grow, is a cornerstone of the argument that PCH is functionally a ribosomopathy, and it should be reproducibly demonstrated if possible.

3. Figure 5: Does depletion of EXOSC8 also result in RPL5-dependent TP53 stabilization? This experiment would give better insight into the phenotypes seen in the zebrafish models.

Minor comments

4. Results:

a. p. 8, end of first paragraph: Please insert the relevant references for MO experiments at "(refs)".

b. p. 8, third paragraph: Perhaps by knocking out *exosc8*, a critical exosome component, nonsense-mediated decay is impaired?

c. p. 9, first paragraph: Please change "3dfp" to "3dpf".

d. p. 16, first paragraph: Please add a reference in the text to Figure 7D.

Reviewer #3 (Comments to the Authors (Required)):

In this article by Müller et al, the authors describe generation of two new homozygous mutant zebrafish lines with mutations in two RNA exosome genes, *exosc8* and *exosc9*. They showed that homozygous mutant larvae had microcephaly, a smaller brain and cerebellum, and defects in pharyngeal arch derived cartilages. Additionally, they performed RNAseq and found an increase in levels of non-coding RNAs in the two models and increased levels of P53 and its targets. The smaller heads of mutants were associated with increased staining of acridine orange, indicating that increased cell death may underlie the microcephaly found. Using human cancer cells, the authors also showed that knockdown of EXOSC9 lead to increased P53 stabilization and G2/M cell cycle arrest. Similarly, P53 transcript levels were also increased in muscle biopsy from patients with mutations in these two genes. However, no significant difference in P53 levels was found in induced neuronal progenitor cells generated from patient fibroblasts, although levels of non-coding RNAs were also significantly affected in these cell lines. This was a straightforward and interesting paper. Experiments performed were well described and overall the conclusions were sound. The authors findings extends upon what had been previously described and provide convincing evidence to suggest that disruption in ribosome biogenesis may contribute to abnormalities associated with mutations in RNA exosome genes.

The major claim of the paper is that RNA exosome associated disorders could be classified as ribosomopathies. This reviewer feels that this claim is substantiated by the RNAseq data in the zebrafish model and to a lesser extent by the data from the cancer cell line. Unfortunately, this assertion was not supported by the iNPC cell lines. Overall, data from the zebrafish model provide strong evidence for disruption in levels of RNA and protein factors important for ribosome biogenesis. Stabilization of P53 and pharyngeal arch defects were also consistent with this hypothesis. However, additional work will be needed to confirm this hypothesis as other unexplored pathways may be responsible for increase of P53 in the cancer cell lines and possibly in the zebrafish model.

Below are some suggestions which may improve the manuscript:

In the introduction, the authors said, " We also performed siRNA-mediated downregulation of EXOSC9 in human cells to confirm the findings from the zebrafish experiments". However, I don't think that one can confirm findings from models in human cell lines. I would suggest rewording this statement.

The authors need to add the missing reference on page 8, first paragraph.

On the top of page 9, dfp should be changed to dpf.

The authors compared the RNAseq data from the exosc8 and exosc9 mutants with those from 3dpf zebrafish exosc3 mutant embryos. However, the authors did not discuss the fact that exosc3 morphants were much more severely affected when compared to exosc8 and exosc9 mutants. Also, for clarity it would be helpful if the authors specify that the RNAseq data used was from morphants injected with the AUG morpholino.

The authors should define hpf the first time that they use it.

The authors reported that they analyzed apoptosis at 48 hpf, in the results and methods sections. however, the embryos in figure 4 are 5dpf according to the figure legend. Hence, the age of the embryos used needs to be clarified.

In the methods sections, the authors should provide some information about the control fibroblast cell lines used to generate iNPC cell lines.

The authors should also discuss if they found any changes in important developmental pathway genes in RNAseq using iNPCs.

For the craniofacial analysis the authors should discuss if the posterior pharyngeal arch derived structures and fins were missing in exosc9 homozygous mutants (Fig 8C).

In the discussion the authors mentioned that clp1 zebrafish morphants were similar to their mutants and died at 5dpf. However, this information was not in the result section.

The discussion could be improved with a short discussion about the molecular and the morphological differences found between exosc8 and exosc9 mutants. Do the authors think that these differences explain differences in patient phenotype? It would also be worth discussing the difference in phenotype found when CRISPR was previously used to generate exosc9 mutant embryos, since the embryos described in this paper seems to be much more mildly affected.

Figure 1: The number of embryos measured should be indicated, the error bars should be defined and the statistical test used should be indicated. In figure 1E the authors should include a loading control for the RT-PCR.

Figure 3 and 4 : The midbrain and cerebellum should be labeled, the number of experiments performed should be indicated in materials and methods section. For figure 4, the number of embryos examined should also be added to the figure legend.

Figure 8: In this figure, the M and CH were very hard to see, maybe a black bold font would be better? Also, these abbreviations should be defined in the figure legend. The error bars should be defined and the statistical method used should be indicated.

Response to reviewers' comments (responses highlighted yellow)

Reviewer #1 (Comments to the Authors (Required)):

In the manuscript by Müller et al. entitled "RNA exosome mutations in models of pontocerebellar hypoplasia alter ribosome biogenesis and p53 levels leading to apoptosis", the authors employ a series of complementary systems and approaches to provide insight into why mutations in genes encoding structural subunits of the RNA exosome cause a spectrum of human diseases. Studies to define the mechanisms that underlie the clinical manifestations in diseases caused by mutations in RNA exosome genes are very important. The types of studies presented here will be critical to provide such insight.

The authors focus primarily on new zebrafish models, but they complement this approach with knockdown in cultured cells as well as induced neuronal progenitor cells (iNPCs) produced from patient fibroblasts. While the authors detect changes in a number of coding and noncoding transcripts, at least one major conclusion highlighted in the title does not seem to hold true in the iNPCs. Comparing different systems is challenging, but these cells are the only that truly model the disease, which is caused by mutant rather than absent proteins so this raises some concerns about a major conclusion of the work.

The RNA exosome both decays and processes RNAs. The direct targets of the RNA exosome are likely to show an increase in response to loss of RNA exosome function. The authors fail to appreciate this mechanistic point in some of their analysis. In fact, the results of the RNA-seq experiment are rather surprising because the authors detected more transcripts that were decreased in the RNA exosome mutant fish than transcripts increased. The transcripts that increase are most likely to be the direct targets of the RNA exosome.

We are not sure why there are more decreased than increased transcripts in the mutant fish. This finding is mainly driven by a larger number of decreased transcripts in the exosc8 mutant fish (supplementary figure 4). Exosome deficiency might act as a negative feedback loop on transcription levels. We have mentioned this on page 10.

Indeed, among the most critical roles of the RNA exosome is certainly rRNA processing to produce the mature rRNA required to assemble functional ribosomes. The authors refer to this point, but do not cite the one study, which demonstrates that mutations that model those found in EXOSC3 cause defects in ribosomal RNA processing (Gillespie A, Gabunilas J, Jen JC, Chanfreau GF. RNA. 2017 Apr;23(4):466-472. Mutations of EXOSC3/Rrp40p associated with neurological diseases impact ribosomal RNA processing functions of the exosome in *S. cerevisiae*. doi: 10.1261/rna.060004.116. Epub 2017 Jan 4. PMID: 28053271). This point supports the authors' supposition and logical suggestion that mutations in the RNA exosome could fall into the category of ribosomopathies.

We have added the missing reference (page 6).

Specific Comments:

In the Introduction and some previous work, the authors state that "In the cytoplasm, the exosome degrades mRNAs that contain AU-rich elements (AREs) and RNAs that have evaded degradation in the nucleus" and reference a review article.

We have added a new reference here and modified the text (page 5).

The authors have generated homozygous crispant/knockout zebrafish to model loss of EXOSC9 and EXOSC8. The authors show sequencing indicating that they have created what should be early nonsense mutations in both genes. They cross these heterozygous fish to produce homozygous fish that should have these non-functional alleles of RNA exosome subunits. The authors analyze these embryos at 5 dpf for most of the subsequent studies. The authors do not provide any information about the viability of these embryos. This point is an important one as no studies have used genetic mutants (rather than depletion) to convincingly show whether the RNA exosome is essential in multicellular organisms. The authors have the opportunity to address this important point.

Information about the viability of the homozygous mutants has been added to the results section, page 9.

With regard to the phenotypes analyzed here, the authors state that 25% of the clutch appear to have smaller head and eyes. The data presented (Figure Legend and/or figure) should include the n values of how many embryos were analyzed to produce the data presented in Figure 1C,D,E.

N values and were added to the figure legend of figure 1.

The authors provide genomic PCR results to confirm the homozygous alleles. While no antibodies are available to confirm the loss of the EXOSC proteins in these fish, the authors need to perform qRT-PCR to evaluate levels of the exosc8 and exosc9 transcripts. An endpoint RT-PCR data presented in the supplemental data shows a decrease in steady-state RNA levels in for exosc9, but the authors state that they detect no difference in the levels of the exosc8 transcript.

As we did not observe reduced exosc8 transcript levels in the RNASeq analysis, we have not performed qPCR to verify the levels of the exosc8 transcript. On the other hand, exosc9 transcript levels were reduced in the RNASeq. We have provided more information of the RNASeq transcript levels on page 9 and 10.

The authors then went on to use these fish to perform RNA-seq analysis. The authors need to clearly state how many independent samples were analyzed for this RNA-seq,

including wildtype or het controls. The methods state that they employed ~30 embryos for each sample- does this constitute a single ensemble biological replicate? At least two replicates should be performed to ensure that differences detected represent real changes rather than variability across the samples.

3 biological replicates (of 30 embryos each) were used for the RNASeq analysis. This has been clarified in the methods section.

Typically for an RNA-Seq analysis, a volcano plot is employed to illustrate the overall changes in transcripts. Given that the RNA exosome is an RNA decay/processing machine, the most direct effects would likely result in an increase in the steady-state levels (via stabilization) of those transcripts. As mentioned above, the finding that more transcripts decrease in both zebrafish models rather than increase raises some concern about whether these changes might reflect indirect effects. The authors need to address this point and speculate about why they see more transcripts decrease than increase.

Volcano plots were added to figure 2 (zebrafish) and figure 6 (iNPCs) and to the supplementary figures. We have addressed possible reasons for decreased transcript levels to page 10.

Figure 2E seems like it could be omitted or moved to the supplemental data.

This figure has been moved to the supplement as suggested.

The authors focus on a modest increase in the level of p53 pathways transcripts. They only validate the result for the p53 transcript in Figure 2G. The data presented in this figure needs to include statistical analysis.

The missing information has been added to the figure and the figure legend.

Figure 5 employs knockdown in cultured human cells to explore the impact of loss of EXOSC9 on p53 levels. The authors' analysis of the zebrafish transcriptome that led them to investigate the increase in p53 levels revealed an increase in the p53 transcript that they cite as a 2.44 fold increase (Figure 2F). This increase is validated by qRT-PCR in Figure 2G (albeit whether this increase is statistically significant is not indicated in this figure).

Statistical analysis has been added to the figure.

In Figure 5A, the authors show that knockdown of EXOSC9 in HCT116 colon cancer cells causes a 3-fold increase in levels of p53 protein. They then state that this could "reflect increased stability and/or higher translation rate of the p53 mRNA in HCT116 cells in the absence of an intact core exosome." The authors need to perform qRT-PCR to examine p53 transcript levels in these knockdown cells. In the zebrafish studies, reagent limitations mean that the authors cannot correlate protein and transcript levels, but here they can and should to provide mechanistic insight into the changes in p53 levels. They could readily examine p53 transcript levels of protein levels under the experimental conditions employed to ascertain the regulatory mechanism. Despite the

result with concomitant knockdown of RPL5- there are multiple other interpretations. The most logical one is- the p53 transcript increases about 3-fold (likely due to increased stability if directly regulated by the RNA exosome, an RNA decay machine) and thus the protein levels also increase 3-fold. An effect on the stability of the p53 protein seems convoluted and perhaps an indirect effects of loss of RNA exosome activity.

We have added new data to the manuscript showing that siRNA-mediated knockdown of EXOSC8 in HCT116 cells also leads to an increase in p53 protein levels. We have expanded the discussion on possible explanation for this increase (page 21). p53 transcript levels have not been analysed in the HCT116 cells. As currently the research laboratories at the universities of Newcastle and Cambridge are closed due to COVID-19, we are not able to perform this experiment.

To reorganize this manuscript without the need for additional experiments, the authors could present Figure 1, Figure 3, Figure 4 (suggesting that apoptosis may occur in small eyes/heads does not require rationale of changes in p53), Figure 7. These studies all characterize the morphology of the zebrafish embryos. The authors could then present the RNA-Seq data in Figure 2 (modifying the presentation to include volcano plots). The Discussion could then mention the logical transcript changes from the RNA-seq that correlate with the morphological changes detected. This approach would eliminate some of the work that absolutely requires additional data to support the conclusions and would also eliminate the iNPCs, which are potentially very exciting, but do not really fit with the narrative here.

This has not been suggested by the journal editors or the other reviewers, therefore we have decided not to restructure and change our manuscript completely by removing all parts referring to the work in human cells.

Minor points:

In the Introduction and some previous work, the authors state that "In the cytoplasm, the exosome degrades mRNAs that contain AU-rich elements (AREs) and RNAs that have evaded degradation in the nucleus" and cite reference 16 (van Hoof A, Lennertz P, and Parker R. Yeast exosome mutants accumulate 3'-extended polyadenylated forms of U4 small nuclear RNA and small nucleolar RNAs. *Molecular and cellular biology*. 2000;20(2):441-52). This reference does not seem to provide any suggestion that targets of the RNA exosome are ARE-containing transcripts and this model is not typically embraced by more recent studies of the RNA exosome. Another study often (and logically) cited to support this model is another somewhat dated study:

Devi Mukherjee 1, Min Gao, J Patrick O'Connor, Reinout Raijmakers, Ger Pruijn, Carol S Lutz, Jeffrey Wilusz. *EMBO J*, 21 (1-2), 165-74 2002 Jan 15 The Mammalian Exosome Mediates the Efficient Degradation of mRNAs That Contain AU-rich Elements

The results reported here likely reflect the fact that ARE-containing transcripts are

often short-lived. Hence, impairing RNA exosome function has an outsized effect in stabilizing transcripts that have a shorter half-life as compared to more stable transcripts. There is little evidence that the RNA exosome complex has specificity for any specific RNA sequences.

The reference has been corrected according to the reviewer's suggestion. We have also modified the text, as the reviewer correctly pointed out, there is no new evidence that ARE-containing transcripts are a preferred target of the exosome (page 5).

The authors refer to "upregulated" genes, a term that has come from the many studies that explore transcriptional changes through RNA-seq analysis. Better terminology here might be to refer to an increase in steady-state levels or just use increase. The term upregulated implies an increase in response to an action- as the RNA exosome decays RNA, the more likely scenario is that loss of RNA exosome function leads to the stabilization of transcripts- rather than upregulation per se. A minor point- but an important one.

The terminology has been modified throughout the manuscript according to the reviewer's suggestion.

Reviewer #2 (Comments to the Authors (Required)):

The exosome is a multipart protein RNA exonuclease complex that functions as an essential regulator of cellular RNA processing and metabolism. To date, multiple genetic variants in core exosome proteins or exosome cofactors have been observed in patients diagnosed with neurodegenerative diseases including pontocerebellar hypoplasia (PCH) and motor neuropathy. Previous work in the field has demonstrated that in patient cells these mutations (1) reduce protein levels, stoichiometrically restricting formation of functional exosome complexes, and (2) result in perturbations of the transcriptome consistent with patient presentations. Additionally, morpholino depletion of these factors in zebrafish has recapitulated muscle weakness and PCH-like phenotypes. However, the full transcriptomic and developmental consequences following complete loss of EXOSC8 or EXOSC9 have yet to be modeled.

In this work, Müller et al. present confirmatory evidence that EXOSC8 and EXOSC9 are critical for normal craniofacial development and transcriptome maintenance by analyzing CRISPR knockout zebrafish. Furthermore, the authors demonstrate that EXOSC9 depletion in human cells stabilizes TP53 in an RPL5-dependent manner, and causes G2/M arrest reminiscent of the nucleolar stress response. Consistent with these observations, transcripts associated with ribosome biogenesis and assembly were significantly upregulated in null EXOSC8 and EXOSC9 zebrafish, possibly as a compensatory mechanism for impaired ribosome biogenesis. Lastly, the authors have shown preliminary data that craniofacial-specific apoptosis, a hallmark of a subset of ribosomopathies, is occurring in the mutant fish. Taken together, Müller et al. argue that diseases like PCH and motor neuropathy arising from mutations in exosome proteins or cofactors can be viewed as ribosomopathies, which is consistent with the importance of the exosome in pre-ribosomal RNA (pre-rRNA) processing.

This manuscript presents both confirmatory and novel evidences for the role of the exosome in development and ribosome biogenesis. In particular, the assertion that diseases caused by exosome mutations are ribosomopathies is valuable and expands the growing list of diseases in which ribosome biogenesis is implicated. However, this claim is mostly predicated on evidence in Figure 2A-B, Figure 4, and Figure 5, which only begin to implicate exosome diseases as ribosomopathies. In particular, the data in Figure 4 is only preliminary and should be replicated before inclusion in published work (see comment 2), and the effect of EXOSC8 depletion on TP53 stabilization is not examined (see comment 3). Additional experiments analyzing defects in pre-rRNA transcription and processing would also strongly support this conclusion, although those could be tabled as future directions. Even more importantly, this manuscript would greatly benefit from an overhaul of its quantitative and statistical data treatments (see comment 1). There are inconsistent explanations for what measures of centrality and error are being graphed, what the sample size for experiments is, and what statistical tests are being performed. These issues should be addressed by the authors before publication.

We are now providing new data of the EXOSC8 knockdown in HCT116 cells. Information to sample size and statistical analysis has been added to the figures. The data in figure 4 shows representative images of wild type and mutant fish embryos. We have analysed 4 different clutches of offspring with 24 embryos each; the acridine orange staining has been performed and evaluated first, followed by genotyping of all embryos that were stained. Image analysis and genotyping have been performed by 2 different researchers (DTB and BM) blinded to each other's results. We therefore believe that the results in figure 4 are not just preliminary.

Major comments

1. Figure 1, 2, 3, 5, 7, S2: Please make clear which measures of centrality and error are being represented graphically, and which (appropriate) statistical tests are being conducted on data in these figures. Figure 5 is the only place any mention of statistical procedures is mentioned. Additionally, t-tests are not appropriate when testing differences between more than two means as in 1C-E, 3B-C, 5A/C, 7B-D, and S2B; please use an appropriate test like one-way ANOVA followed by a post-hoc test.

Information to sample size and statistical analysis has been added to the figures. T-tests have been performed to compare each mutant separately to the wild type (exosc8 to wt and exosc9 to wt); the exosc8 mutant and the exosc9 mutant have not been compared to each other. The t-test is a valid test for comparing 2 groups.

2. Figure 4: The claim that KO fish exhibit increased apoptosis would be strengthened by including quantification of the acridine orange staining across multiple fish. As-is, n = 1 for each condition is preliminary and is not adequate to strongly make this claim, and cannot reveal if EXOSC8 or EXOSC9 is more important for this hallmark phenotype of ribosomopathies. The observation of upregulated craniofacial-specific apoptosis, rather

than simple failure of those cells to grow, is a cornerstone of the argument that PCH is functionally a ribosomopathy, and it should be reproducibly demonstrated if possible.

As outlined above, the data in figure 4 shows representative images of wild type and mutant fish embryos. We have analysed 4 different clutches of offspring with 24 embryos each; the acridine orange staining has been performed and evaluated first, followed by genotyping of all embryos that were stained. Image analysis and genotyping have been performed by 2 different researchers (DTB and BM) blinded to each other's results. We therefore believe that the results in figure 4 are not just preliminary.

3. Figure 5: Does depletion of EXOSC8 also result in RPL5-dependent TP53 stabilization? This experiment would give better insight into the phenotypes seen in the zebrafish models.

We have added new data to figure 5 showing that a depletion of EXOSC8 also leads to an increase in p53 levels.

Minor comments

4. Results:

a. p. 8, end of first paragraph: Please insert the relevant references for MO experiments at "(refs)".

b. p. 8, third paragraph: Perhaps by knocking out *exosc8*, a critical exosome component, nonsense-mediated decay is impaired?

c. p. 9, first paragraph: Please change "3dfp" to "3dpf".

d. p. 16, first paragraph: Please add a reference in the text to Figure 7D.

We have corrected all of these as required.

Reviewer #3 (Comments to the Authors (Required)):

In this article by Müller et al, the authors describe generation of two new homozygous mutant zebrafish lines with mutations in two RNA exosome genes, *exosc8* and *exosc9*. They showed that homozygous mutant larvae had microcephaly, a smaller brain and cerebellum, and defects in pharyngeal arch derived cartilages. Additionally, they performed RNAseq and found an increase in levels of non-coding RNAs in the two models and increased levels of P53 and its targets. The smaller heads of mutants were associated with increased staining of acridine orange, indicating that increased cell death may underlie the microcephaly found. Using human cancer cells, the authors also showed that knockdown of EXOSC9 lead to increased P53 stabilization and G2/M cell cycle arrest. Similarly, P53 transcript levels were also increased in muscle biopsy from patients with mutations in these two genes. However, no significant difference in P53 levels was found in induced neuronal progenitor cells generated from patient

fibroblasts, although levels of non-coding RNAs were also significantly affected in these cell lines. This was a straightforward and interesting paper. Experiments performed were well described and overall the conclusions were sound. The authors findings extends upon what had been previously described and provide convincing evidence to suggest that disruption in ribosome biogenesis may contribute to abnormalities associated with mutations in RNA exosome genes.

The major claim of the paper is that RNA exosome associated disorders could be classified as ribosomopathies. This reviewer feels that this claim is substantiated by the RNAseq data in the zebrafish model and to a lesser extent by the data from the cancer cell line. Unfortunately, this assertion was not supported by the iNPC cell lines. Overall, data from the zebrafish model provide strong evidence for disruption in levels of RNA and protein factors important for ribosome biogenesis. Stabilization of P53 and pharyngeal arch defects were also consistent with this hypothesis. However, additional work will be needed to confirm this hypothesis as other unexplored pathways may be responsible for increase of P53 in the cancer cell lines and possibly in the zebrafish model.

Below are some suggestions which may improve the manuscript:

In the introduction, the authors said, "We also performed siRNA-mediated downregulation of EXOSC9 in human cells to confirm the findings from the zebrafish experiments". However, I don't think that one can confirm findings from models in human cell lines. I would suggest rewording this statement.

This statement has been reworded.

The authors need to add the missing reference on page 8, first paragraph.

This reference has been added.

On the top of page 9, dfp should be changed to dpf.

This has been corrected.

The authors compared the RNAseq data from the exosc8 and exosc9 mutants with those from 3dpf zebrafish exosc3 mutant embryos. However, the authors did not discuss the fact that exosc3 morphants were much more severely affected when compared to exosc8 and exosc9 mutants. Also, for clarity it would be helpful if the authors specify that the RNAseq data used was from morphants injected with the AUG morpholino.

We added a sentence to clarify this as suggested.

The authors should define hpf the first time that they use it.

The abbreviation hpf has been defined on page 11.

The authors reported that they analyzed apoptosis at 48 hpf, in the results and methods sections. However, the embryos in figure 4 are 5dpf according to the figure legend. Hence, the age of the embryos used needs to be clarified.

The acridine orange staining for apoptosis has been performed at 48hpf. The mistake in the figure legend has been corrected.

In the methods sections, the authors should provide some information about the control fibroblast cell lines used to generate iNPC cell lines.

This information has been added to the method section on page 26.

The authors should also discuss if they found any changes in important developmental pathway genes in RNAseq using iNPCs.

We have added information about changes found in developmental pathway genes in the iNPCs to page 16.

For the craniofacial analysis the authors should discuss if the posterior pharyngeal arch derived structures and fins were missing in exosc9 homozygous mutants (Fig 7C).

We only measured ceratohyal and Mackel's cartilage in our mutant fish, but the appearance of the more posterior pharyngeal cartilage is different in the homozygous mutant fish, possibly also influenced by the change in the more anterior cartilage structures. However, we did not analyse this in detail. We did not notice that the fins were missing in the mutants.

In the discussion the authors mentioned that clp1 zebrafish morphants were similar to their mutants and died at 5dpf. However, this information was not in the result section.

The information was added to the results section on page 8.

The discussion could be improved with a short discussion about the molecular and the morphological differences found between exosc8 and exosc9 mutants. Do the authors think that these differences explain differences in patient phenotype? It would also be worth discussing the difference in phenotype found when CRISPR was previously used to generate exosc9 mutant embryos, since the embryos described in this paper seem to be much more mildly affected.

The head and brain size measurements suggest that exosc9 mutant fish have a slightly milder phenotype than exosc8 mutants. Neuromuscular junctions however, are more affected in exosc9 mutants (supplementary data) and both mutants die at the same developmental stage. Currently it is difficult to compare disease severity in patients with EXOSC8 mutations versus EXOSC9 mutations as there is only a very small number of patients reported. It might be possible that EXOSC9 patients have milder clinical symptoms than EXOSC8 patients, but it may also depend on the exact mutation. For

EXOSC3 mutations, where many more patients have been reported, a clearer genotype-phenotype comparison has shown that certain EXOSC3 mutations are associated with milder symptoms and better survival rates.

The *exosc8* and *exosc9* mutant fish also have a slightly milder phenotype than our previously published *exosc8* morphants (Boczonadi et al, 2014) and the *exosc9* morphants and crispants (Burns et al, 2018). For morpholino-mediated knockdown, studies have shown that the morphant phenotype is more severe than the genetic mutant phenotype, because there are some compensatory effects by other genes that become upregulated and rescue the phenotype (Rossi et al, 2015). This might also be the case for the crispants, although it has not been studied yet. Some F0 injected crispants could show a more severe phenotype due to the injection process of RNA and Cas9. Furthermore, F0 fish might contain mutations in other genes caused by the guideRNA binding to other genomic sites; those are outcrossed and removed in later generations. However, we emphasize that also in our earlier study (Burns et al, 2018), *exosc9* crispants have a predominantly mild phenotype (only 16% have a moderate or severe phenotype).

We have added a new paragraph to the discussion.

Figure 1: The number of embryos measured should be indicated, the error bars should be defined and the statistical test used should be indicated. In figure 1E the authors should include a loading control for the RT-PCR.

The information has been added to figure 1.

Figure 3 and 4 : The midbrain and cerebellum should be labeled, the number of experiments performed should be indicated in materials and methods section. For figure 4, the number of embryos examined should also be added to the figure legend.

We have changed the figures and their legends as suggested.

Figure : In this figure, the M and CH were very hard to see, maybe a black bold font would be better? Also, these abbreviations should be defined in the figure legend. The error bars should be defined and the statistical method used should be indicated.

The labelling and the missing information have been added to the figure and the figure legend.

May 22, 2020

RE: Life Science Alliance Manuscript #LSA-2020-00678-TR

Prof. Rita Horvath
University of Cambridge
Department of Clinical Neurosciences
University Neurology Unit, Level 5 'A' Block, Box 165
Cambridge Biomedical Campus
Cambridge CB2 0QQ
United Kingdom

Dear Dr. Horvath,

Thank you for submitting your revised manuscript entitled "RNA exosome mutations in pontocerebellar hypoplasia alter ribosome biogenesis and p53 levels". As you will see, the reviewers appreciate the introduced changes, and we would thus be happy to publish your paper in Life Science Alliance pending final minor revisions:

- Please address the remaining reviewer concern
- Please note that all figures need to adhere to our figure guidelines; Fig. 2 & Fig. 6, as well as Fig. S1, Fig. S4, Fig. S5, Fig. S6, span currently multiple pages, which we cannot accept for publication here
- Please add scale bars in Fig 1 A,B, Fig. 3. Fig. 4, Fig. S5A-F, Fig. 7A, Fig. S2, Fig. S3
- please upload also the supplementary figures as individual files; the legends should remain in the main manuscript docx file
- Please add callouts in the text to Fig 1A, Fig. 3A, Fig. S1A-D, Fig. S4A-C, Fig. S5A-G, Fig. S6, S7, S8
- Table S5, S6, S7 are missing from the submission, please upload
- Please also deposit the raw RNA-seq data in repository and provide the accession code in your manuscript file

A. FINAL FILES:

B. MANUSCRIPT ORGANIZATION AND FORMATTING:

Thank you for your attention to these final processing requirements.

Sincerely,

Andrea Leibfried, PhD
Executive Editor
Meyerhofstr. 1
69117 Heidelberg, Germany

t +49 6221 8891 414
e contact@life-science-alliance.org
www.life-science-alliance.org

Reviewer #1 (Comments to the Authors (Required)):

The authors have addressed most of the points raised in the previous critiques. Some data cannot be collected due to the lab shutdown, but the data presented are sufficient to support the authors conclusions.

Reviewer #2 (Comments to the Authors (Required)):

This manuscript presents both confirmatory and novel evidences for the role of the exosome in development and ribosome biogenesis. In particular, the assertion that diseases caused by exosome mutations are ribosomopathies is valuable and expands the growing list of diseases in which ribosome biogenesis is implicated. Most claims made by this manuscript are strongly supported by the data presented and are suitable for publication. However, more comprehensive data needs to be included for Figure 4, which suggests that EXOSC8 and EXOSC9 knockout fish may exhibit increased craniofacial apoptosis. Per their methods, the authors have likely analyzed enough fish to strongly support this claim, but only show one representative image per condition in Figure 4. A graphical quantification summarizing all data relevant to Figure 4 is in order. For example, the authors might graph the average total acridine orange staining per fish for each genotype. They could also graph the percentage of fish with increased AO staining over the WT baseline for each EXOSC mutant. No additional experiments would be required to create this graph, although its inclusion would greatly strengthen claims based on Figure 4.

Dear Dr. Leibfried, dear Editors,

We wish to thank you for provisionally accepting our manuscript entitled "RNA exosome mutations in pontocerebellar hypoplasia alter ribosome biogenesis and p53 levels" for publication in your journal. We have addressed the remaining points raised by the editors and by reviewer 2. In detail, we have made the following changes:

- We have added the quantification of the acridine orange staining to figure 4 as new figure 4B as requested.
- We have modified the figures that were spanning multiple pages in the main manuscript and the supplement. Scale bars have been added to Fig. 1 A,B, Fig. 3. Fig. 4, Fig. S5A-F, Fig. 7A, Fig. S2, and Fig. S3. We uploaded the supplementary figures as individual files and added the legends to the main manuscript.
- We added callouts in the text to Fig. 1A, Fig. 3A, Fig. S1A-D, Fig. S4A-C, Fig. S5A-G, Fig. S6, S7, S8. The supplementary tables were now uploaded individually.
- We deposited the raw RNA-seq data to the NCBI Gene Expression Omnibus (GEO) and provided the accession number in the methods section.

The changes made to the manuscript text are highlighted in yellow.

We hope that our manuscript is now fully acceptable for publication in *Life Science Alliance*.

Please don't hesitate to contact us if you need further information.

Sincerely,

Rita Horvath

June 3, 2020

RE: Life Science Alliance Manuscript #LSA-2020-00678-TRR

Prof. Rita Horvath
University of Cambridge
Department of Clinical Neurosciences
University Neurology Unit, Level 5 'A' Block, Box 165
Cambridge Biomedical Campus
Cambridge CB2 0QQ
United Kingdom

Dear Dr. Horvath,

Thank you for submitting your Research Article entitled "RNA exosome mutations in pontocerebellar hypoplasia alter ribosome biogenesis and p53 levels". It is a pleasure to let you know that your manuscript is now accepted for publication in Life Science Alliance. Congratulations on this interesting work.

*****IMPORTANT:** If you will be unreachable at any time, please provide us with the email address of an alternate author. Failure to respond to routine queries may lead to unavoidable delays in publication.*******

DISTRIBUTION OF MATERIALS:

Again, congratulations on a very nice paper. I hope you found the review process to be constructive and are pleased with how the manuscript was handled editorially. We look forward to future exciting

submissions from your lab.

Sincerely,

Reilly Lorenz
Editorial Office Life Science Alliance
Meyrhofstr. 1
69117 Heidelberg, Germany
t +49 6221 8891 414
e contact@life-science-alliance.org
www.life-science-alliance.org